# Preparation of Cellulose-Grafted Acrylic Acid Stabilized Jujube Branch Biochar-Supported Nano Zero-Valent Iron Composite for Cr(VI) Removal from Water

**DOI:** 10.3390/nano15060441

**Published:** 2025-03-14

**Authors:** Xiaoxue Wang, Zhe Tan, Shuang Shi, Shanyuan Zhang, Shuang Yang, Xingyu Zhang, Pingqiang Gao, Yan Zhang

**Affiliations:** 1School of Chemistry and Chemical Engineering, Yulin University, No. 51 Chongwen Road, Yulin 719000, China; 2School of Materials Science and Engineering, Xi’an University of Technology, No. 5 Jinhua South Road, Beilin District, Xi’an 710048, China; 3Yulin Engineering Research Center of Coal Chemical Wastewater, Yulin University, No. 51 Chongwen Road, Yulin 719000, China

**Keywords:** jujube branch, biochar, nZVI, Cell-g-PAA, removal, Cr(VI)

## Abstract

A stabilized biochar (BC)–nano-scale zero-valent iron (nZVI) composite (BC-nZVI@Cell-g-PAA) was prepared using cellulose-grafted polyacrylic acid (Cell-g-PAA) as the raw material through in situ polymerization and liquid-phase reduction methods for the remediation of hexavalent chromium (Cr(VI))-contaminated water. BC-nZVI@Cell-g-PAA was characterized by XRD, FT-IR, SEM, BET, TEM, and XPS. According to the batch experiments, under optimized conditions (Cr(VI) concentration of 50 mg/L, pH = 3, and dosage of 2 g/L), the BC-nZVI@Cell-g-PAA composite achieved maximum Cr(VI) removal efficiency (99.69%) within 120 min. Notably, BC, as a carrier, achieved a high dispersion of nZVI through its porous structure, effectively preventing particle agglomeration and improving reaction activity. Simultaneously, the functional groups on the surface of Cell-g-PAA provided excellent protection for nZVI, significantly suppressing its oxidative deactivation. Furthermore, the composite effectively reduced Cr(VI) to insoluble trivalent chromium(Cr(III)) species and stabilized them on its surface through immobilization. The synergistic effects of physical adsorption and chemical reduction greatly contributed to the removal efficiency of Cr(VI). Remarkably, the composite exhibited excellent reusability with a removal efficiency of 62.4% after five cycles, demonstrating its potential as a promising material for remediating Cr(VI)-contaminated water. In conclusion, the BC-nZVI@Cell-g-PAA composite not only demonstrated remarkable efficiency in Cr(VI) removal but also showcased its potential for practical applications in environmental remediation, as evidenced by its sustained performance over multiple reuse cycles. Moreover, Cr(VI), a toxic and carcinogenic substance, poses significant risks to aquatic ecosystems and human health, underscoring the importance of developing effective methods for its removal from contaminated water.

## 1. Introduction

With the rapid expansion of industrial activities, the contamination of water bodies becomes severe because more and more harmful heavy metals are released into the aquatic environment. Chromium (Cr) is one of the heavy metals in the water body environment that has received significant attention based on its extensive usage in various industrial fields, such as mining, electroplating, steel manufacturing, and dye production. Industrial emissions of Cr are a major source of environmental pollution [1], with compounds of hexavalent Cr(VI) being widely recognized for their strong carcinogenic and genotoxic characteristics, posing significant risks to human health. Hexavalent chromium (Cr(VI)) and trivalent chromium (Cr(III)) are the two main oxidation states in Cr contamination [2]. The former has been given increasing attention according to its great mobility and substantial harmfulness [3], and therefore, requires urgent and effective mitigation strategies.

Nano-scale zero-valent iron (nZVI) is a propitious remedy for Cr(VI)-polluted water based on its large surface area, great efficiency, and strong reducing capability [4]. However, its practical application is constrained by inherent challenges, such as particle aggregation and susceptibility to oxidation, which impact its efficacy. To overcome these challenges [5], studies have long explored various strategies and carriers, such as bentonite, mesoporous silica, kaolin, zeolite, activated carbon, and biochar (BC), to stabilize nZVI and reduce particle aggregation [6]. Among these, BC, a highly porous and carbon-rich material, is obtained through the thermal decomposition of biomass under high temperatures and oxygen-limited environments. Its outstanding physicochemical characteristics include a customizable surface area, well-defined porous structures, and a high density of oxygen-containing functional groups [7]. BC is distinguished by its extensive surface area, structural stability, and potent adsorption capabilities. In addition, the oxygen-containing functional groups abundant on the surface of BC can enhance its ability to adsorb heavy metal ions [8]. The jujube tree planting density in the Yulin region is high, yet the discarded jujube branches are typically burned or landfilled, which poses ecological risks. By converting these branches into functional materials, the regional waste biomass can be effectively utilized. Therefore, BC, due to its porous structure and rich cellulose content, has gained increasing interest as a precursor from discarded jujube branches. Converting this agricultural waste into BC through pyrolysis can allow effective usage of agricultural waste products and offer a sustainable solution in order to treat Cr-contaminated water bodies [9]. However, the limited specific surface area, underdeveloped pore structure, and simplistic surface chemistry of pristine biochar constrain its performance in practical applications. To address these limitations, chemical activation has become a key strategy for enhancing biochar properties. Among various activation methods, KOH activation is widely utilized in biochar modification research due to its efficiency and practicality. KOH activation not only significantly increases the specific surface area and porosity of biochar, optimizing its physical structure, but also introduces abundant oxygen-containing functional groups, thereby improving its surface chemistry [10]. This leads to a substantial enhancement of its adsorption capacity for heavy metal ions.

The surface coating technique is another method available for the improvement of the dispersibility and the stability of nZVI [11]. Various polymer materials, including starch, chitosan, and carboxymethyl cellulose (CMC), have been employed to create protective coatings on nZVI particles to improve their water treatment performance [12]. The interaction potential energy between nano iron particles primarily includes van der Waals attraction potential energy, double-layer potential energy, and magnetic attraction potential energy. The presence of coating materials increases the double-layer repulsion potential energy and increases the spatial potential energy between nano iron particles, collectively counteracting the magnetic attraction potential energy, thereby reducing the aggregation of nano iron and enhancing its stability [13].

Cellulose-grafted polyacrylic acid (Cell-g-PAA) contains abundant polar functional groups, e.g., carboxyl and hydroxyl. It not only precisely anchors onto the nanoparticle surface in a bidentate bridging manner but also, in excess, can form a gel-like network structure through hydrogen bonding, entanglement, and cross-linking interactions. This network structure is highly favorable for binding and encapsulating nZVI. Furthermore, Cell-g-PAA is non-toxic, biodegradable [14], and exhibits excellent adsorption capabilities for heavy metals [15]. Therefore, Cell-g-PAA is a promising candidate for surface coating.

Cr(VI) is a highly carcinogenic and mutagenic heavy metal that poses serious threats to human health and the environment. Due to its high solubility and strong oxidative properties, Cr(VI) easily spreads in water bodies, endangering aquatic life and potentially affecting human health through the food chain. Prolonged exposure can lead to skin diseases, respiratory disorders, and cancers [16]. The pollution caused by Cr(VI) not only harms ecosystems but also threatens the safety of agricultural irrigation and drinking water, necessitating the urgent development of efficient and sustainable remediation methods. The aim of this research was to investigate how Cell-g-PAA encapsulated BC-nZVI@Cell-g-PAA composite to improve the removal of Cr(VI) from water. Through harnessing the stability of BC and the protective attributes of Cell-g-PAA, the issue of nZVI particle aggregation could be tackled in order to enhance dispersal and stability. The outcomes of this study hold significant importance in formulating effective strategies for Cr pollution control in aquatic environments.

## 2. Experiment

### 2.1. Chemicals and Equipment

The jujube branches, obtained from the Jiaxian area in Yulin, China, were thoroughly washed, dried at 105 °C, and subsequently crushed using a grinder. The obtained powder was sieved with a 150-mesh screen. The chemical content of the jujube branch powder was tested using an organic elemental analyzer (OEA; UNICUBE, Dresden, Germany), revealing the elemental composition to be 47.21% of carbon (C), 5.90% of hydrogen (H), 41.35% of O, and 0.42% of nitrogen (N) (Table 1).

Chemical reagents, namely sodium hydroxide (NaOH), N,N-methylene bisacrylamide cross-linking agent (C_7_H_10_N_2_O_2_), sulfuric acid (H_2_SO_4_), and potassium hydroxide (KOH) were obtained from Tianjin Kemiou Chemical Reagent Co., Ltd. (Tianjin, China). Sodium borohydride (NaBH_4_), n-hexane (CH_3_(CH_2_)_4_CH_3_), potassium persulfate initiator (K_2_S_2_O_8_) and diphenylcarbazide (C_13_H_14_N_4_O) were acquired from China National Pharmaceutical Group Chemical Reagent Co., Ltd. (Beijing, China). Acetone (CH_3_COCH_3_) was bought from Sichuan Xilong Chemical Co., Ltd. (Beijing, China). Cellulose (C_6_H_10_O_5_)_n_ was obtained from Hefei BASF Biotechnology Co., Ltd. (Hefei, China). Polyglycerol fatty acid ester (PGFE) was acquired from Zhengzhou Dahe Food Technology Co., Ltd. (Zhengzhou, China). Acrylic acid (C_3_H_4_O_2_) was bought from Tianjin DaMao Chemical Reagent Factory (Tianjin, China). Ferrous sulfate (FeSO_4_) was procured from Tianjin Sheng’ao Chemical Reagent Co., Ltd. (Tianjin, China). Phosphoric acid (H_3_PO_4_) was obtained from Tianjin Tianli Chemical Reagent Co., Ltd. (Tianjin, China). Potassium dichromate (K_2_Cr_2_O_7_) was sourced from Tianjin Ruijinte Chemicals Co., Ltd. (Tianjin, China).

### 2.2. Sample Synthesis

#### 2.2.1. Preparation of BC and BC-nZVI

The preparation of BC was conducted through high-temperature carbonization of the powdered jujube branch. The branch powder, sourced from the Jiaxian area in Yulin, China, was subjected to low-temperature pyrolysis and high-temperature activation in a laboratory-scale tube furnace (XL-RL, Yangzhou Xingliu Electric Appliance Co., Ltd., Yangzhou, China). The process entailed heating the powder at 5 °C/min in a nitrogen atmosphere, maintained at 600 °C for 2 h to produce BC. The obtained BC was mixed with KOH (with a mass ratio of BC:KOH = 1:2), a ratio determined after comparing multiple sets of preliminary experiments, and then heated at 800 °C for 2 h in a tube furnace to obtain KOH-activated BC. KOH activation increases the porosity, surface functional groups, and specific surface area of the biochar, thereby enhancing its adsorption capacity and ability to remove pollutants. After activation, the mixture was naturally cooled and stirred in a 1 mol/L HCl solution for 4 h. Finally, it was washed until the pH was neutral and dried at 105 °C.

In this study, a BC-nZVI composite material with a carbon-to-iron ratio (C:Fe) of 1:1 was prepared. Specifically, 0.50 g of BC was dissolved in 2.48 g of FeSO_4_·7H_2_O and placed in a beaker. The mixture was then subjected to ultrasonic treatment to ensure thorough dispersion of the biochar. Subsequently, NaBH_4_ solution was slowly added dropwise to the mixture until bubbling ceased, indicating the completion of the reaction. Finally, the product was washed twice with ultrapure water to obtain the BC-nZVI sample.

#### 2.2.2. Synthesis of BC-nZVI@Cell-g-PAA Composites

A 50 mL aqueous solution containing 4.8 g of cellulose in a 500 mL flask was prepared using mechanical stirring at 90 °C for 30 min. The above flask was equipped with a reflux condenser, stirrer, thermometer, and nitrogen line. When the mixture was quenched to 35–40 °C, 100 mL n-hexane and 0.12 g of Polyglycerol esters of fatty acids (PGFE) (a suspending stabilizer) were added and the flask agitated for 15 min. After that, potassium persulfate (K_2_S_2_O_8_) initiator was introduced and the solution was further stirred for 20 min before partially neutralized acrylic acid added. Then, 2.5 mL of N,N-methylene bisacrylamide cross-linking agent (NN) (5 g/L) was added. The polymerization procession was carried out at 70 °C for 3 h. The BC-nZVI powder was then mixed with Cell-g-PAA solution (the mass ratio of BC to Cell-g-PAA was 2:1). This product was stirred at ambient temperature for 30 min and then dried using freeze-drying equipment at −60 °C for 8 h under vacuum. The preparation flowchart of the BC-nZVI@Cell-g-PAA composite material is shown in Figure 1.

### 2.3. Characterization Methods

XRD patterns were obtained from a D8 X-ray diffractometer (Bruker, Ettlingen, Germany). This instrument features a copper (Cu) target X-ray source and a Lynx detector array. IR spectra of the samples were recorded on a TENSOR 27 FTIR Spectrometer (Bruker, Germany). The morphology of the samples was studied using a Sigma 300 Field Emission Scanning Electron Microscope (ZEISS, Jena, Germany) with an Oxford X-Max Extreme SDD energy-dispersive X-ray spectroscopy (EDS) detector. An Escalab 250Xi X-ray Photoelectron Spectroscope (XPS) (Thermo, Waltham, MA, USA) was employed to analyze the surface functional groups and elemental content of composite materials (before and after reaction with Cr(VI), respectively). HRTEM was performed using a J Tecnai G2 F30 Transmission Electron Microscope (FEI, Hillsboro, OR, USA). The specific surface area (SSA) was determined with an ASAP 2460 physical adsorption analyzer (Micromeritics, Norcross, GA, USA).

### 2.4. Cr(VI) Removal Experiments

Batch experiments were performed in a 250 mL conical flask on a reciprocating shaker at 25 °C with a rotary speed of 150 rpm. The preparation of Cr(VI) solution involves dissolving K_2_Cr_2_O_7_ in distilled water to prepare a stock solution of 1000 mg/L Cr(VI). Prior to use, the stock solution is diluted with deionized water, and the initial pH of the solution is adjusted using 0.1 mol/L NaOH and 0.1 mol/L HNO_3_. Typically, 2 g/L BC-nZVI@Cell-g-PAA was added to 100 mL of 50 mg/L Cr(VI) solution. At specific time intervals, 1 mL reaction samples were obtained by filtration, and examined immediately using a UV spectrophotometer (Shimazu UV-2450, Shimadzu, Kyoto, Japan). The concentration of Cr(VI) is determined using the diphenylcarbazide spectrophotometric method [17]. The principle is that diphenylcarbazide acts as a color reagent, which reacts with Cr(VI) under acidic conditions to form a purple-red compound. Under acidic conditions, Cr(VI) reacts with diphenylcarbazide to form diphenylcarbazone, while the reduced Cr(III) product forms a complex with diphenylcarbazone, producing a purple-red substance. This product has a maximum absorption at a wavelength of 540 nm, and the absorbance is linearly related to the Cr(VI) concentration within a certain range. The specific experimental procedure is as follows: (1) Filter the sample and transfer 1 mL of the supernatant into a 50 mL colorimetric tube, then dilute to the mark. (2) Add 0.5 mL of sulfuric acid and phosphoric acid (1:1), followed by 2 mL of the color reagent solution, and shake to mix. (3) After standing for 5–10 min, measure the absorbance using a 3 cm cuvette, with water as the reference, at a wavelength of 540 nm using a UV spectrophotometer. This allows determination of the Cr(VI) concentration. The Cr(VI) removal rate, *R* (%), was calculated from Equation (1). All the batch experiments were repeated in triplicate, and an average was obtained. In the single-factor experiments, we used the error bar analysis method to process the data. This approach visually presents the degree of data dispersion and uncertainty, making the analysis of experimental results more scientific and rigorous.(1)$$R=\frac{C_{0}-C_{t}}{C_{0}}\times 100\%,$$
where *C*_0_ and *C_t_* are the initial Cr(VI) concentration and Cr(VI) concentration at time *t*, respectively.

### 2.5. Isotherm and Adsorption Kinetics

To clarify the Cr(VI) removal mechanism of the BC-nZVI@Cell-g-PAA composite material, adsorption isotherms and adsorption kinetics models were employed to fit the experimental data. Within the scope of adsorption isotherm models, the Langmuir and Freundlich models, which are among the most widely used classical models, were applied to explain the equilibrium distribution relationship of the solute between the solid and liquid phases during the adsorption process. In terms of adsorption kinetics models, the pseudo-first-order (PFO) and pseudo-second-order (PSO) models were selected to analyze the adsorption rate changes over time, further elucidating the rate-controlling steps and the reaction mechanisms of the adsorption process.

The fitting equations of the two adsorption isotherm models can be expressed as follows [18]:(2)$$\frac{C_{e}}{q_{e}}=\frac{C_{e}}{q_{m}}+\frac{1}{q_{m}K_{L}}$$(3)$$\log q_{e}=\log K_{F}+\frac{1}{n}\log C_{e}$$
where $C_{e}$ (mg/L) is the equilibrium concentration of Cr(VI); $K_{L}$ (L/mg) is the equilibrium constant of the Langmuir isotherm model; $K_{F}$ (mg/g) is the equilibrium constant of the Freundlich isotherm model; and 1/*n* represents the adsorption intensity, such that *n* < 1 indicates high adsorption difficulty, 1 < *n* < 2 indicates moderate adsorption difficulty, and 2 < *n* < 10 indicates high adsorption intensity and relatively easy adsorption [19].

The fitting equations of the two adsorption kinetics models can be expressed as follows [20]:(4)$$\ln\left(q_{e}-q_{t}\right)=\ln q_{e}-k_{1}t,$$(5)$$\frac{t}{q_{t}}=\frac{1}{k_{2}q_{e}^{2}}+\frac{t}{q_{e}},$$
where $q_{t}$ (mg/g) signifies the adsorption capability at time *t*, $q_{e}$ (mg/g) is the adsorption capability at equilibrium, and $k_{1}$ (min^−1^) and $k_{2}$ (g·mg^−1^·min^−1^) represent the rate constants for the pseudo-first- and pseudo-second-order adsorption kinetics.

## 3. Results and Discussion

### 3.1. Characterization of Samples

Figure 2 shows the SEM photographs of the samples. As shown in Figure 2a, BC has a rough and porous surface, which provides favorable conditions for the introduction of nZVI and the preparation of BC-nZVI composites. From Figure 2b, the nZVI spheres are aggregated in a chain-like formation. In Figure 2c, nZVI is uniformly dispersed on the surface of BC, providing more adsorption reaction sites for hexavalent chromium. Additionally, minimal agglomeration of nZVI occurs on the BC surface, with some particles forming larger aggregates. A portion of nZVI is also attached to the pores of BC, effectively immobilizing the nZVI during the reaction process and preventing its loss. For the BC-nZVI@Cell-g-PAA composite, numerous flocculent objects are found on the surface (Figure 2e).

Additionally, a homogeneous dispersion of the nZVI particles is evenly distributed on the surface of the composite, which is of benefit to improve Cr(VI) removal using the BC-nZVI@Cell-g-PAA composite prepared through the adsorption-encapsulation method [21]. Furthermore, encapsulated by Cell-g-PAA, the antioxidative property of the nZVI is removed (Figure 2e,f), revealed by needle-like structures covering the composite surface after the reaction. This is the result of the continuous redox reaction of Fe^0^ within the composite material to generate Fe(OH)_2_, Fe(OH)_3_, and Cr(OH)_3_ precipitates that cover the surface during adsorption [22].

In order to understand further the surface elements, the EDS spectra of the different samples were obtained and analyzed (Figure 3). The surfaces of BC-nZVI and BC-nZVI@Cell-g-PAA composite were principally comprised of C, O, and Fe (Figure 3a,b). Cr was found (Figure 3c), suggesting that Cr was adsorbed and immobilized on the surface of the BC-nZVI@Cell-g-PAA composite after reaction with the Cr-contaminated solution (Figure 3c) [23]. Furthermore, the EDS images indicate that the Fe content increased after the Cell-g-PAA coating, suggesting that the coating layer effectively isolates oxygen and water from the external environment, thereby reducing Fe oxidation. The coating layer serves to protect nZVI, further confirming the critical role of the Cell-g-PAA coating in enhancing nZVI performance. This finding provides strong support for the optimization of material design aimed at the removal of heavy metal pollutants [24].

HRTEM provides an intuitive way to observe the distribution of Fe particles on the BC-nZVI@Cell-g-PAA composite surface, along with key characteristics, such as particle uniformity, size, crystal facet exposure, and interplanar spacing (Figure 4). The dispersal of Fe nanoparticles on the BC surface was uniform and the average size of these particles was 9.8 nm. This uniform dispersion of Fe relied on the relatively large surface area of BC. Furthermore, the lattice spacing of the (110) and (200) crystal facet positions were 0.246 and 0.273 nm, respectively [25]. The thickness of the Cell-g-PAA coating layer affects the Cr(VI) removal performance. Specifically, as the Cell-g-PAA coating ratio increases, the reduction ability of Cr(VI) first improves and then decreases. The improvement is attributed to two main reasons: (1) The larger the molecular weight of Cell-g-PAA, the stronger the electrostatic repulsion effect it provides, leading to smaller and more stable nZVI particles with a higher surface area, thus enhancing reactivity; and (2) Cell-g-PAA effectively isolates nZVI from oxygen, preventing premature oxidation. However, when the coating layer becomes too thick, it significantly increases the viscosity of the nZVI slurry, causing a reduction in the reduction ability, as well as difficulties in transfer and drying. The relevant image is shown in Appendix A.

Figure 5a displays the FT-IR absorption spectra of BC, BC-nZVI, BC-nZVI@Cell-g-PAA, and RBC-nZVI-PAA. For BC, the peaks observed in the spectrum are primarily associated with the functional groups of the biochar matrix. In the BC-nZVI spectrum, the main absorption peaks appear at 3430 cm^−1^ (–OH stretching vibration), 1620 cm^−1^ (C=C stretching vibration), 1720 cm^−1^ (C=O stretching vibration), and 1160 cm^−1^ (C–O bending vibration) [26]. As for the BC-nZVI@Cell-g-PAA composite material, the intensities of the peaks at 3430 cm^−1^ (–OH) and 1720 cm^−1^ (C=O) are significantly higher than those of BC, indicating an increase in oxygen-containing functional groups, which could enhance the adsorption of Cr(VI). Specifically, functional groups such as C=O and –OH can form stable complexes with Cr(VI), thereby improving the material’s ability to adsorb Cr(VI). This is consistent with findings from previous studies [27] that showed oxygen-containing functional groups enhance the material’s hydrophilicity and iron chelation capacity. In particular, when dealing with Cr(VI)-contaminated water, the presence of oxygen-containing functional groups can accelerate the removal process of Cr(VI). Therefore, it can be inferred that BC-nZVI@Cell-g-PAA has more O-based functional groups [28], which can accelerate the rapid adsorption of Cr(VI) ions from the polluted solution. In addition, the peak intensities of phenolic-OH and C-O for BC-nZVI@Cell-g-PAA decrease after reacting with Cr(VI) [29]. The complex formed by phenolic-OH and Cr(VI) acts as an electron shuttle, facilitating the electron transfer. This electron transfer mechanism facilitates the reduction process of Cr(VI), further enhancing the removal efficiency of Cr(VI). Similar mechanisms have been validated in other studies, particularly when oxygen-containing functional groups in materials interact with Cr(VI). These functional groups not only participate in the adsorption process but also promote the occurrence of electron transfer reactions, which is crucial for improving the overall Cr(VI) removal performance [30].

Figure 5b presents the XRD patterns of all samples (BC, nZVI, BC-nZVI, BC-nZVI@Cell-g-PAA, and RBC-nZVI@Cell-g-PAA). For BC, two strong diffraction peaks emerge at 2θ = 26.0° and 43.1° that correspond to the (002) and (100) crystal planes of the graphite structure, respectively, showing that there is graphite component in biochar. This observation is consistent with previous studies, which have also identified similar diffraction peaks in biochar derived from various biomass sources [31]. A characteristic peak exhibited at 44.6° in nZVI, BC-nZVI, and BC-nZVI@Cell-g-PAA indicates the presence of metallic Fe^0^, which was confirmed to be nZVI through comparison with Power Diffraction File (PDF) standard cards (International Center for Diffraction Data). Compared to nZVI, the peak from Fe^0^ in BC-nZVI becomes weak as BC is introduced; it further decreases for BC-nZVI@Cell-g-PAA due to encapsulation of the composite by the Cell-g-PAA polymer material. This phenomenon has also been reported in similar studies, where polymer coatings or carriers have been demonstrated to reduce the crystallinity of nZVI [32]. Moreover, new diffraction peaks appeared in the BC-nZVI@Cell-g-PAA XRD pattern at 2θ = 14.1°, 27.0°, 36.2°, and 46.7°. These peaks are associated with FeO(OH) and correspond to the (020), (120), (031), and (051) crystal planes in the PDF card. This indicates that Fe^0^ has undergone a redox reaction and has been converted to FeO(OH) [33]. This is consistent with the findings reported in [34]. This observation corroborates the HRTEM and SEM results, further confirming the presence of metal Fe crystal facets in the composite.

The XPS study was employed to explore the interaction between the BC-nZVI@Cell-g-PAA and Cr(VI). As shown in Figure 6a, the characteristic peaks for O 1s, Fe 2p, and Cr 2p were detected, confirming BC-nZVI@Cell-g-PAA to be primarily composed of C, O, and Fe. Consistent with SEM and EDS results, it was evident that Cr 2p was present on the surface of BC-nZVI@Cell-g-PAA after the removal processes [35]. Subsequently, the composition ratios and states of these elements were investigated by narrow scans performed for O 1s, Fe 2p, and Cr 2p.

From the XPS spectrum in Figure 6b, three peaks are observed for the O 1s signal, with peak centers at 530.1 eV, 531.7 eV, and 533.2 eV, corresponding to the characteristic peaks from oxides (Fe-O), surface hydroxyl groups (C-O/OH), and adsorbed water (H_2_O), respectively. After reacting with Cr(VI), FeO(OH) was formed. It can be seen from the figure that the intensities of the phenolic-OH and C-O peaks in the BC-nZVI@Cell-g-PAA composite after reacting with Cr(VI) are weakened, particularly the phenolic-OH. As reported by Chen et al. [25], after complexing with Cr(VI), the Mulliken charge of the aromatic ring significantly increases, which allows for electrostatic interaction and adsorption of Cr(VI). This is also consistent with the previous FT-IR analysis.

As shown in Figure 6c and Table 2, peaks at 711.2 and 724.3 eV correspond to Fe 2p_3/2_ and Fe 2p_1/2_, respectively. In addition, the spikes at 713.7 and 726.8 eV are indicative of Fe(III). Weak characteristic peaks at 707.3 and 720.3 eV correspond to Fe^0^. The peaks at 718.8 and 729.5 eV correspond to satellite peaks. Before reacting with Cr(VI), the surface of BC-nZVI@Cell-g-PAA consisted of 5.1% Fe^0^, 78.2% Fe(II), and 16.7% Fe(III). The proportion of Fe^0^ reduced to 0%, Fe(II) reduced to 64.1%, and Fe(III) increased to 35.9% after the reaction. This result indicates Fe^0^ and Fe(II) are the primary species that reduce Cr(VI). This is consistent with previous studies. The literature [36] indicates that Fe^0^ and Fe(II) play a key role in the reduction of Cr(VI) to Cr(III) through an electron transfer mechanism. The oxidation of Fe^0^ and Fe(II) to Fe(III) further supports this reduction process.

As in Figure 6d, two peaks at 577.3 and 586.7 eV are attributed to Cr(III) 2p_1/2_ and Cr(III) 2p_3/2_, respectively. Moreover, the peaks at 580.0 and 589.3 eV correspond to Cr(VI) 2p_3/2_ and Cr(VI) 2p_1/2_, respectively. Among these, the proportions of Cr(III) and Cr(VI) were 85.1% and 14.9%, respectively. The simultaneous presence of Cr(III) and Cr(VI) on the surface of BC-nZVI@Cell-g-PAA after the reaction suggested that the removal processes included both physical adsorption of Cr(VI) onto the composite surface and its chemical reduction to Cr(III). Chemical reduction is the predominant mechanism for Cr(VI) removal according to the larger proportion of Cr(III). This is in agreement with the findings of [37]. Similar studies have also emphasized the importance of chemical reduction in the removal of Cr(VI), particularly in systems involving nZVI-based composites.

Figure 7 presents the N_2_ adsorption–desorption isotherms and pore size distribution curves of four different materials. As shown in Figure 7a, the desorption curve of the original BC is nearly a straight line, indicating a limited pore structure. The adsorption–desorption curve of KOH-activated BC conforms to a Type I isotherm, with a rapid increase in adsorption at low relative pressure and an average pore size of 1.72 nm, suggesting a well-developed microporous structure [38]. However, in the relative pressure range of P/P₀ = 0.6–0.9, the increase in adsorption slows down, indicating the presence of a small amount of mesopores. Similarly, BC-nZVI also follows a Type I isotherm, with an average pore size of 2.23 nm. Compared with the original BC, BC-nZVI exhibits a larger adsorption volume, although it remains smaller than that of KOH-activated BC. This could be attributed to the introduction of nZVI, which occupies part of the pores and thus restricts further expansion of the pore structure. BC-nZVI@Cell-g-PAA also follows a Type I isotherm, with an average pore size of 2.38 nm. However, its adsorption volume is smaller than that of BC-nZVI, possibly due to the modification by Cell-g-PAA, which limits the openness of the pore structure and results in a lower adsorption capacity compared to BC-nZVI and KOH-activated BC.

Table 3 shows the pore structure parameters of the material. The specific surface area of BC is 1987.6 m^2^/g; after the loading of nZVI, the specific surface area of BC-nZVI decreased to 735.6 m^2^/g. This reduction is likely due to the deposition of nZVI particles on the surface of BC during the liquid-phase reduction process, filling or adhering to the pore structures of the BC, which significantly reduces the material’s specific surface area. Furthermore, partial oxidation of nZVI may occur during subsequent storage or usage, forming iron oxides (such as Fe_2_O_3_ and Fe_3_O_4_). These oxides further adhere to the pore surfaces or inside the pore channels of BC, narrowing the pores or even completely blocking them, thereby further decreasing the specific surface area. Furthermore, after the addition of the coating layer, the specific surface area of the BC-nZVI@Cell-g-PAA composite material decreased further to 142.3 m^2^/g. This reduction is likely due to the introduction of Cell-g-PAA, which altered the surface morphology of BC-nZV [39]. The originally rough and porous surface was covered by a denser polymer layer, which might have infiltrated the internal pores of BC-nZVI, reducing the pore volume and thereby further decreasing the specific surface area.

### 3.2. Optimal Conditions for Cr(VI) Reduction

As shown in Figure 7a, with increasing doses of the materials, the Cr(VI) removal rate increased gradually. This is because when the dosage of BC-nZVI@Cell-g-PAA increased, the number of reaction sites increased as well as the absorption surface, which improved the interaction possibilities between BC-nZVI@Cell-g-PAA and Cr(VI) [40]. However, when the dosage was increased to 2 g/L, the removal efficiency of Cr(VI) did not significantly improve despite the further increase in material usage. The reason is that under conditions where the Cr(VI) solution has constant concentration the number of contaminants is fixed. Despite increasing the dosage of BC-nZVI@Cell-g-PAA, the removal rate tends to stabilize. Furthermore, the order of Cr(VI) removal for the different materials was as follows: BC < nZVI < BC-nZVI < BC-nZVI@Cell-g-PAA. Therefore, to enhance in situ reduction efficiency while maintaining high effectiveness and cost-effectiveness, the optimal dosage selected is 2 g/L for the BC-nZVI@Cell-g-PAA composite material.

To more accurately simulate the actual pH environment of wastewater, we conducted experimental studies on the removal of Cr(VI) using composite materials under different pH conditions. From Figure 8b, the Cr(VI) removal rate of BC-nZVI@Cell-g-PAA gradually decreased with increased pH, indicating that acidic conditions are more conducive for Cr(VI) elimination than basic conditions. Many studies have shown that Cr(VI) in different ionic forms can be found in water solutions, including CrO_4_^2−^, Cr_2_O_7_^2−^, Cr_3_O_10_^2−^, Cr_4_O_13_^2−^, and HCrO_4_^−^ [41]. Under acidic conditions, the primary state of Cr(VI) is HCrO_4_^−^, whereas CrO_4_^2−^ begins to dominate when pH > 6. H^+^ ions adsorbed onto the surface of the composite conduct a positive charge at low pH conditions. Simultaneously, Cr(VI) ions present in different anionic forms are strongly attracted to the positively charged adsorbent surface, further enhancing the Cr(VI) removal efficiency. Additionally, the passivating layer of Fe(III)-Cr(III) hydroxide compounds on the surface of BC-nZVI@Cell-g-PAA leads to a reduction in the degree of corrosion of the nZVI-based material. Therefore, such conditions can better expose effective active sites [31]. On the flip side, a substantial quantity of OH^−^ ions exists in alkaline solutions, which results in a negatively charged adsorbent surface. The subsequent repulsions between BC-nZVI@Cell-g-PAA and Cr(VI) anions lead to reduced adsorption efficiency [42].

The Cr(VI) removal process by BC-nZVI@Cell-g-PAA reached equilibrium within 30 min as shown in Figure 8c. As the composite came into contact with the Cr(VI) solution, Cr(VI) competed with the positive charges on the BC-nZVI@Cell-g-PAA surface. This led to a swift decline in the Cr(VI) concentration, subsequently diminishing the driving force for mass transfer [43]. With increasing contact time, the active ion exchange sites and internal pores of the composite surface gradually fill with Cr(VI) ions. Simultaneously, as time progresses, the adsorbed anions gradually aggregate to form complexes, hindering further diffusion and leading to the gradual attainment of dynamic adsorption equilibrium. When BC-nZVI@Cell-g-PAA was added to the adsorption reaction, in the middle to late stages (30–120 min), the increasing adsorption time reduced the concentration of Cr(VI). Because of the precipitation of Fe and Cr, the passivating layer on the composite surface was formed. The passivating layer impedes effective contact between the Cr(VI) and the composite particles, obstructing the removal processes, and further affecting the increase in Cr(VI) removal rate [44].

The removal rate significantly decreased when the initial Cr(VI) concentration was raised. This reason is that the number of available active sites on the BC-nZVI@Cell-g-PAA surface is constant (Figure 8d). When the Cr(VI) concentration reaches a threshold, it occupies all available active sites, saturating the composite material surface [45]. Simultaneously, the increasing Cr(VI) concentration, on the one hand, leads to an increased collision frequency between Cr(VI) and nZVI, and, on other hand, produces a higher likelihood of nZVI oxidation, resulting in the loss of its reduction capability. This redox reaction generates a passivating layer of Fe(III)-Cr(III), impeding the continuation of the reaction as previously described. As a result of the increasing Cr(VI) concentration, the passivating layer was formed rapidly, which reduced the Cr(VI) removal rate [46].

When the temperature was increased, the Cr(VI) removal efficiency was enhanced (Figure 8d), which indicates the Cr(VI) adsorbed by BC-nZVI@Cell-g-PAA is involved in an endothermic process, and raising the adsorption temperature can be conducive to reaching the activation energy required for the overall chemical reaction, thereby promoting Cr(VI) adsorption [47]. Furthermore, when the temperature of BC-nZVI@Cell-g-PAA increased from 15 °C to 35 °C, the Cr(VI) removal efficiency increased from 91.62% to 99.95%. The reason is that the mass transfer of Fe ions in the solution is faster at higher temperatures, making the particle move quickly and promoting the faster transfer of Cr(VI) to the BC-nZVI@Cell-g-PAA surface, allowing efficient Cr(VI) removal.

In appraising the reusability of the composite, BC-nZVI@Cell-g-PAA was studied for five consecutive cycles with the same initial experimental conditions. There was a certain loss of removal performance: the removal rate was 99% for the first use and 62.4% after the fifth use (Figure 8f). An amount of BC-nZVI@Cell-g-PAA was taken out for each sampling. BC-nZVI@Cell-g-PAA was treated with alkali solution and acid solution after each repeated experiment, and an amount of BC-nZVI@Cell-g-PAA was lost in the cleaning process [48]. The result suggested that BC-nZVI@Cell-g-PAA has a relatively stable reusability for its real utilization.

### 3.3. Cr(VI) Adsorption Isotherms

The Cr(VI) adsorption capacity of BC-nZVI@Cell-g-PAA at a temperature of 25 °C, BC-nZVI@Cell-g-PAA dosage of 2 g/L, and reaction time of 2 h was investigated by two different isotherm models, the Langmuir isotherm model and Freundlich adsorption isotherm model. The fitting results are shown in Figure 9 and Table 4.

The *R^2^* value for the Langmuir adsorption model exceeded that of the Freundlich model. This suggests a closer fit of the adsorption process to the Langmuir model, namely monolayer adsorption. Additionally, the 1/n value being less than 1 suggested that the adsorption process was chemical and spontaneous in nature [49].

### 3.4. Cr(VI) Adsorption Kinetics

At a temperature of 25 °C, initial Cr(VI) concentration of 50 mg/L, and BC-nZVI@Cell-g-PAA dosage of 2 g/L, the removal kinetics of Cr(VI) were analyzed using two different models, the pseudo-first-order kinetic model and the pseudo-second-order kinetic model, respectively, to investigate the adsorption and reduction characteristics of BC-nZVI@Cell-g-PAA towards Cr(VI).

The experimental data was fitted according to Equations (4) and (5) (Figure 10) and the corresponding parameters are shown in Table 5. For BC-nZVI@Cell-g-PAA, the *R^2^* value for the pseudo-first-order kinetic linear fitting for was 0.87, while that for the pseudo-second-order kinetic linear fitting was 0.99, suggesting a stronger fit for the latter. This indicated that the removal process was strongly associated with the adsorption sites on the surface of BC-nZVI@Cell-g-PAA and that chemical adsorption played a dominant role in the reaction process [50].

### 3.5. Cr(VI) Reduction Mechanism

As previously mentioned, Cr(VI) exists predominantly as HCrO_4_^−^ at 1 < pH < 6 and as CrO_4_^2−^ at pH > 6, and acidic conditions are more advantageous to eliminate Cr(VI) (Figure 11). The primary Cr(VI) removal processes of BC-nZVI@Cell-g-PAA includes adsorption, reduction, and co-precipitation. Firstly, BC possesses a porous structure and a large SSA, providing ample surface adsorption sites. Secondly, under acidic conditions, positively charged BC-nZVI@Cell-g-PAA can effectively adsorb and enhance the aggregation of Cr(VI) anions onto its surface. Thirdly, within BC-nZVI@Cell-g-PAA, nZVI and divalent Fe ions (Fe^0^ and Fe (II)) serve as electron donors, effectively promoting the reduction of Cr (VI) to Cr (III), as expressed in the following equations [51]:2HCrO_4_^−^ + 3Fe^0^ + 14H^+^ → 2Cr^3+^ + 3Fe^2+^ + 8H_2_O,(6)HCrO_4_^−^ + 3Fe^2+^ + 7H^+^ → Cr^3+^ + 3Fe^3+^ + 4H_2_O,(7)2CrO_4_^2−^ + 3Fe^0^ + 8H_2_O → 2Cr^3+^ + 3Fe^2+^ + 16OH^−^,(8)CrO_4_^2−^ + 3Fe^2+^ + 4H_2_O → Cr^3+^ + 3Fe^3+^ + 8OH^−^,(9)

Lastly, as the reaction progresses, the OH^−^ radicals generated on the BC-nZVI@Cell-g-PAA surface lead to an increase in the system pH, causing precipitation of Cr(III) hydroxides or mixed Fe(II)/Cr(III) (oxy)hydroxides, which are removed from the solution, as expressed in the following equations:(1 − *x*)Fe^3+^ + *x*Cr^3+^ + 3H_2_O → Cr*_x_*Fe_1−*x*_(OH)_3_ + 3H^+^,(10)(1 − *x*)Fe^3+^ + *x*Cr^3+^ + 2H_2_O → Cr*_x_*Fe_1−*x*_OOH + 3H^+^,(11)

This also explains why acidic conditions are conducive to Cr(VI) removal as low pH conditions speed up the corrosion of nZVI, which enhances the Cr(VI) and Fe^0^/Fe(II) reaction rate.

## 4. Conclusions

A BC-nZVI@Cell-g-PAA composite material was synthesized using in situ polymerization and liquid-phase reduction methods. Through optimization, it was determined that under conditions of a Cr(VI) concentration of 50 mg/L, pH of 3, and a dosage of 2 g/L, the BC-nZVI@Cell-g-PAA composite achieved the highest Cr(VI) removal efficiency of 99.69% within 120 min. This composite material exhibits several key characteristics: on the one hand, BC provides a large surface area, and nZVI particles are uniformly distributed on the BC surface. On the other hand, Cell-g-PAA offers excellent protection, enhancing the dispersion and stability of the composite. Furthermore, the principal mechanism for eliminating Cr(VI) using BC-nZVI@Cell-g-PAA entailed (i) the conversion of Fe^0^ to Fe(II) and Fe(III), (ii) reduction of Cr(VI) to Cr(III), and (iii) Cr removal via adsorption, reduction, and co-precipitation. The reaction process involved the synergistic action of physical adsorption and chemical reduction of Cr(VI) based on a pseudo-second-order kinetic model. Finally, the inclusion of a Cell-g-PAA polymer layer on the surface of nZVI increased the number of O-based functional groups, further enhancing the efficacy of Cr(VI) removal.

This description of the BC-nZVI@Cell-PAA material, with its efficient, stable, and environmentally friendly properties, provides scientific evidence and technical support for the practical application of nano zero-valent iron (nZVI) in the removal of Cr(VI). It demonstrates broad application potential. The pseudo-second-order kinetic model indicated that the removal process was strongly associated with the adsorption sites on the surface of BC-nZVI@Cell-g-PAA and that chemical adsorption played a dominant role in the reaction proceess. Additionally, the Langmuir model showed the monolayer adsorption of the BC-nZVI@Cell-g-PAA composite towards (Cr(VI)) happened onto a surface containing a finite number of identical sites. Most important, the absorbed BC-nZVI@Cell-g-PAA composite can be regenerated and recycled. Furthermore, the reduction of Cr(VI) demonstrates significant specificity. In terms of environmental conditions, the reduction process demands specific parameters, with the reduction rate of Cr(VI) varying across different pH levels. Typically, reduction to Cr(III) occurs more readily under acidic conditions, whereas the reaction may be inhibited under alkaline conditions. Regarding the impact of the products, Cr(VI) is highly toxic and mobile, while the reduced Cr(III) is less toxic and tends to precipitate, leading to a substantial decrease in environmental mobility and ecological toxicity. This transformation from high toxicity to low toxicity highlights the specific role of Cr(VI) reduction in areas such as environmental remediation. These results demonstrate that the unique BC-nZVI@Cell-g-PAA exhibits great potential for application in water treatment.

## Figures and Tables

**Figure 1 nanomaterials-15-00441-f001:**
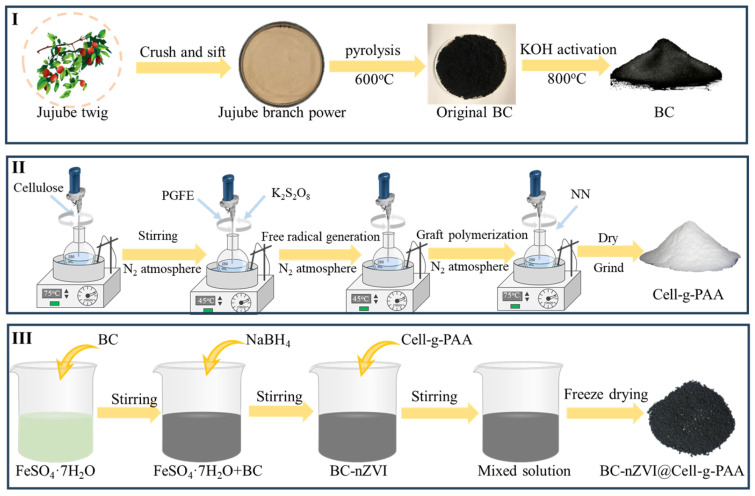
Schematic diagram of the preparation process of BC-nZVI@Cell-g-PAA composite (I: Preparation of BC; II: Preparation of Cell-g-PAA; III: BC-nZVI@Cell-g-PAA preparation of composite materials).

**Figure 2 nanomaterials-15-00441-f002:**
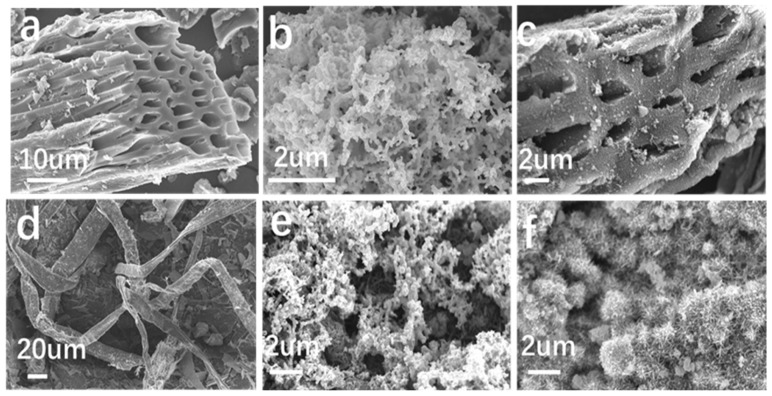
The SEM (**a**–**f**) images of BC, nZVI, BC-nZVI, Cell-g-PAA, BC-nZVI@Cell-g-PAA, and BC-nZVI@Cell-g-PAA after Cr(VI) removal.

**Figure 3 nanomaterials-15-00441-f003:**
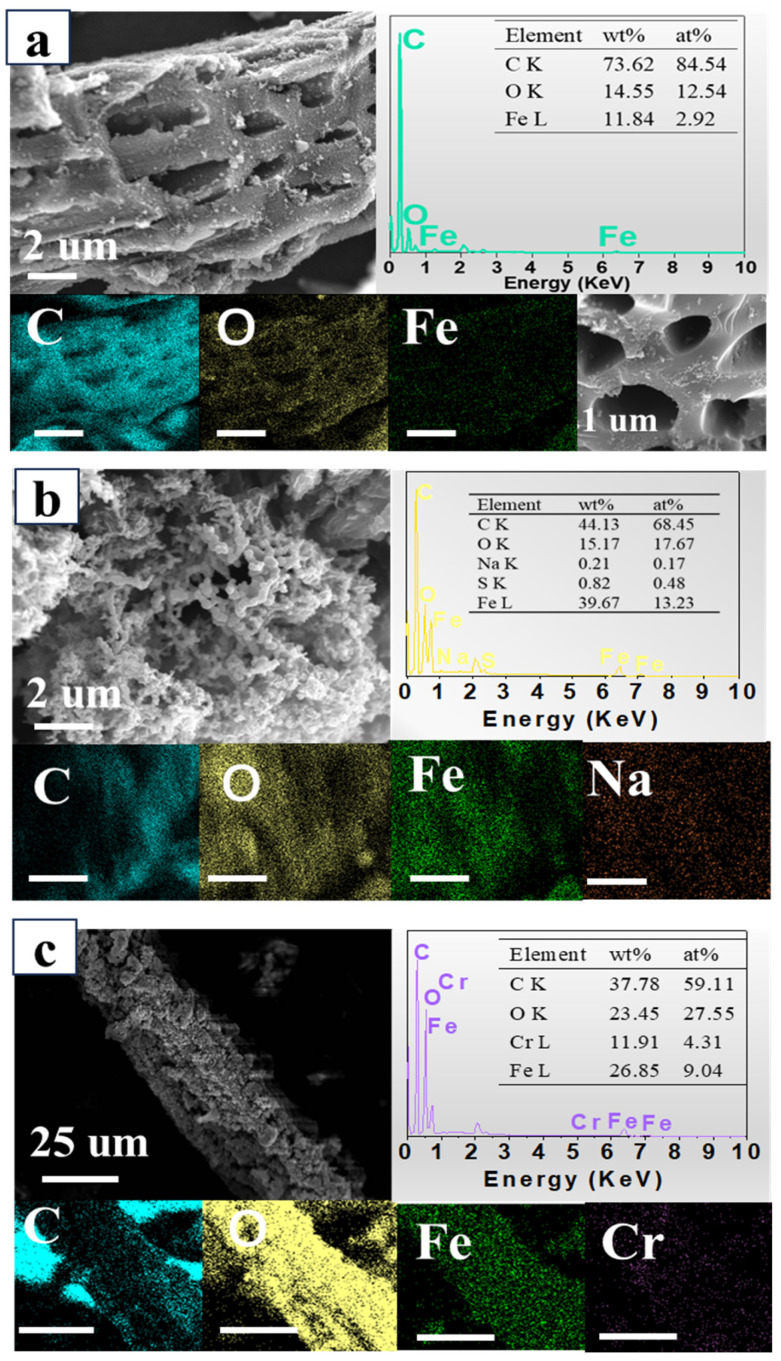
The SEM images and EDS element mapping of (**a**) BC-nZVI, (**b**) BC-nZVI@Cell-g-PAA, and (**c**) RBC-nZVI@Cell-g-PAA.

**Figure 4 nanomaterials-15-00441-f004:**
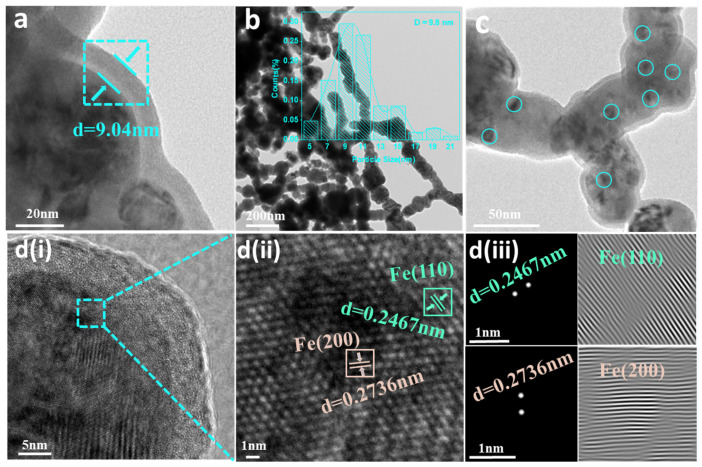
(**a**) HRTEM thickness analysis of the Cell-g-PAA coating layer of the composite; (**b**) Fe particle size distribution; (**c**) Fe particle distribution on the composite surface; and (**d**) (**i**) HRTEM images, (**ii**) lattice spacing, and (**iii**) electron interval diffraction patterns of Fe in the composite.

**Figure 5 nanomaterials-15-00441-f005:**
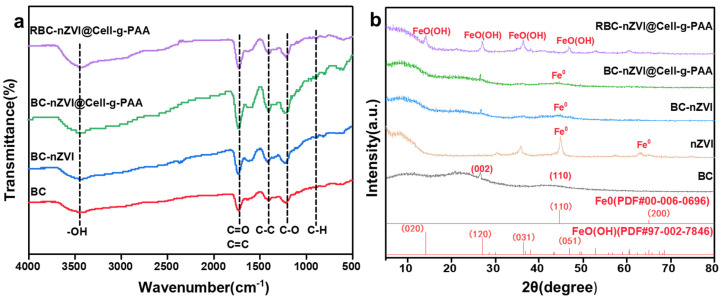
(**a**) FT-IR and (**b**) XRD patterns of different samples.

**Figure 6 nanomaterials-15-00441-f006:**
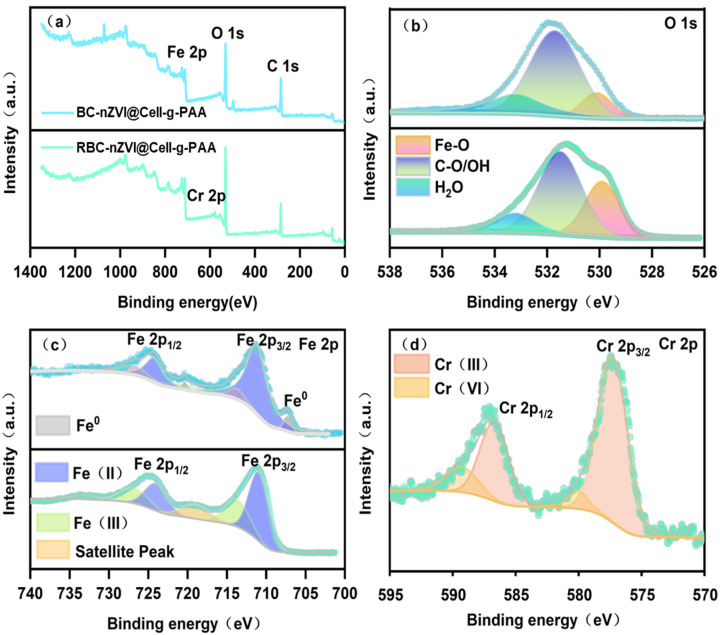
(**a**) Wide-scan XPS spectra, and high-resolution XPS spectra of BC-nZVI@Cell-g-PAA and RBC-nZVI@Cell-g-PAA; (**b**) O 1s; (**c**) Fe 2p; and (**d**) Cr 2p.

**Figure 7 nanomaterials-15-00441-f007:**
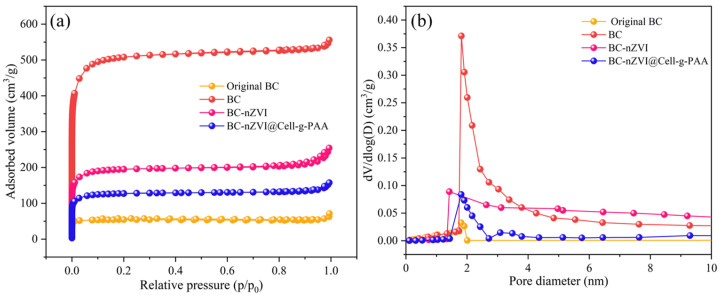
(**a**) N_2_ adsorption–desorption isotherm diagram; (**b**) aperture profile.

**Figure 8 nanomaterials-15-00441-f008:**
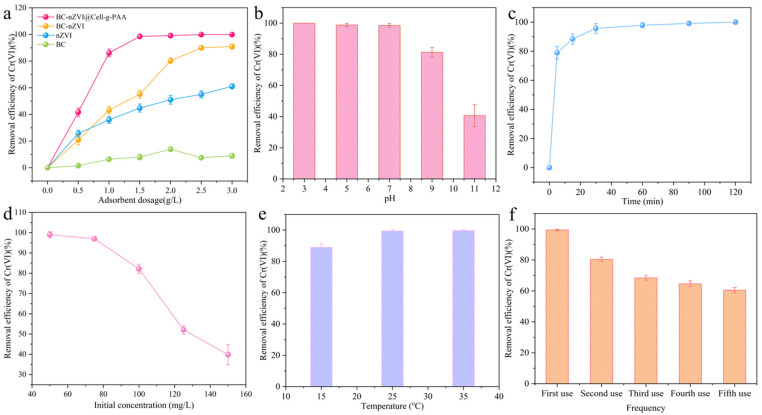
Influence of (**a**) adsorbent dosage, (**b**) pH, (**c**) time, (**d**) initial concentration, (**e**) temperature, and (**f**) frequency on Cr(VI) removal by BC-nZVI@Cell-g-PAA.

**Figure 9 nanomaterials-15-00441-f009:**
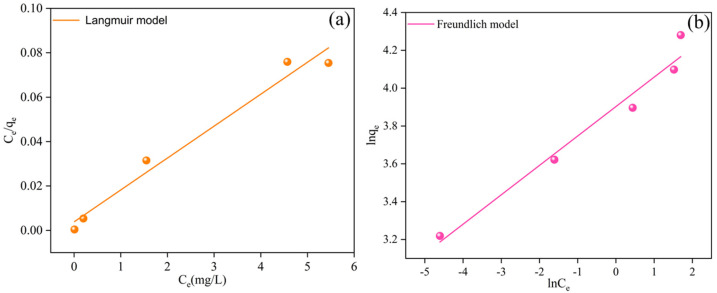
(**a**) Langmuir and (**b**) Freundlich adsorption isotherms.

**Figure 10 nanomaterials-15-00441-f010:**
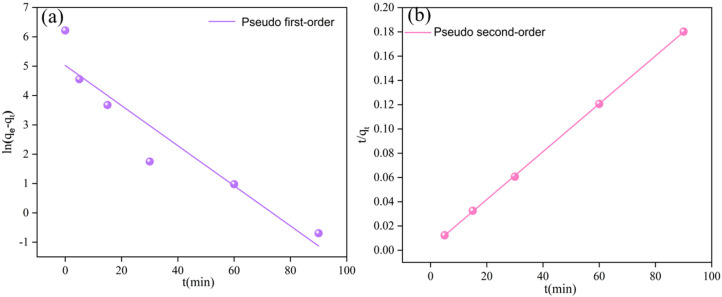
Soprtion kinetics of Cr(VI) onto BC-nZVI@Cell-g-PAA (**a**) First-order kinetic model, (**b**) Second-order dynamic model.

**Figure 11 nanomaterials-15-00441-f011:**
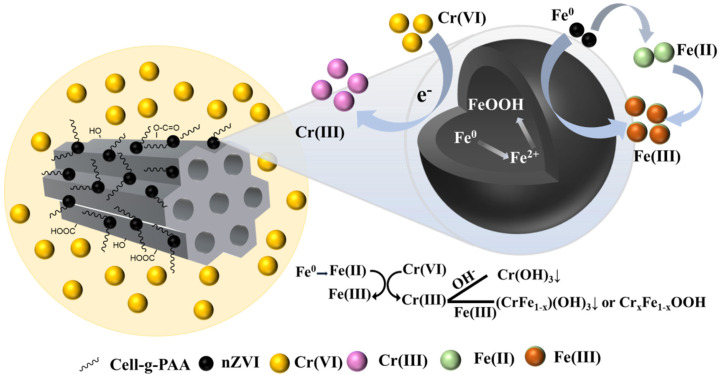
Cr(VI) removal mechanism of BC-nZVI@Cell-g-PAA.

**Table 1 nanomaterials-15-00441-t001:** Jujube branch chemical composition.

Chemical Component	C	H	O	N	Other Metals
Content (%)	47.21	5.90	41.35	0.42	5.10

**Table 2 nanomaterials-15-00441-t002:** Photoelectron spectra peak area, binding energy (BE), and half-peak width (FWHM) for O, Fe, and Cr on the composite. Here, B and A refer to BC-nZVI@Cell-g-PAA and RBC-nZVI@Cell-g-PAA.

Species BE (eV)		FWHM(eV) ^a^		Area (%)		Species
		B	A	B	A	
	530.1	1.3	1.5	11.8	30.2	Fe-O
O 1s	531.7	2.2	1.9	65.7	55.7	C-O/OH
	533.2	2.3	1.7	22.5	14.1	H_2_O
	707.3/720.3	1.2/0.7	/	5.1	0.0	Fe^0^
Fe 2p	711.2/724.3	3.1/2.6	2.9/3.3	78.2	64.1	Fe(II)
	713.7/726.8	2.3/3.5	3.7/4.2	16.7	35.9	Fe(III)
	577.3/586.7	/	2.8/3.7	/	85.1	Cr(III)
Cr 2p	580/589.3	/	1.8/3.1	/	14.9	Cr(VI)

^a^ Peak width at half height.

**Table 3 nanomaterials-15-00441-t003:** The specific surface area and pore structure parameters of original BC, BC, BC-nZVI, and BC-nZVI@Cell-g-PAA composites.

Sample	SSA (m^2^ g^−1^)	Pore Volume (m^3^ g^−1^)		Average Pore Size (nm)
		*V* _total_	*V* _mic_	
Original BC	220.87	0.1	0.08	1.79
BC	1987.6	0.86	0.66	1.72
BC-nZVI	735.6	0.39	0.25	2.23
BC-nZVI@Cell-g-PAA	142.3	0.07	0.12	2.38

**Table 4 nanomaterials-15-00441-t004:** Parameters of isotherm models for Cr(VI) adsorption by BC-nZVI@Cell-g-PAA.

Sample	Langmuir Model			Freundlich Model		
	q_m_ (mg/g)	K_L_ (L/mg)	R^2^	K_F_ (mg/g)	1/n	R^2^
BC-nZVI@Cell-g-PAA	69.54	3.75	0.96	49.50	0.15	0.95

**Table 5 nanomaterials-15-00441-t005:** Pseudo-first- and pseudo-second-order kinetics fitting parameters.

Model	Parameter	Sample
Pseudo-first-order	K_1_ (min^−1^)	0.06
	R^2^	0.87
Pseudo-second-order	K_2_·(g·mg^−1^·min^−1^)	0.01
	R^2^	0.99

## Data Availability

Data are contained within the article and Appendix A.

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
