# Peer review of "Preparation of Cellulose-Grafted Acrylic Acid Stabilized Jujube Branch Biochar-Supported Nano Zero-Valent Iron Composite for Cr(VI) Removal from Water"

_nanomaterials, 2025, doi:10.3390/nano15060441_

Round 1
Reviewer 1 Report
Comments and Suggestions for Authors
Dear Editor of Nanomateroials MDPI, thank you for including me as part of your reviewing team and improving the quality of submitted manuscripts. I have thoroughly reviewed the manuscript entitled " Preparation of cellulose-grafted acrylic acid stabilized biochar-nano-scale zero-valent iron composite for Cr(VI) removal from water" and consider it suitable for your journal, after the authors make the following adjustments and respond point by point to the comments.
Comments to authors
- In the abstract section, include a sentence at the end that exemplifies the conclusion
- In the introductory section, include the following information to reinforce the definition of biochar: "Biochar, a highly porous and carbon-rich material, is obtained through the thermal decomposition of biomass under high temperatures and oxygen-limited environments. Its outstanding physicochemical characteristics, such as customizable surface area, well-defined porous structures, and a high density of oxygen-containing functional groups." Citation to: (2024). Valorization of Agave Angustifolia Bagasse Biomass from the Bacanora Industry in Sonora, Mexico as a Biochar Material: Preparation, Characterization, and Potential Application in Ibuprofen Removal. Sustainable Chemistry, 5(3), 196-214.
- In the materials and methods section I recommend making an image that represents the design of Synthesis of "BC-nZVI@Cell-g-PAA composites", you can do it separately or in the same figure 1.
- In the materials and methods section, include a section that specifies how the absorption isotherms were carried out, it can be within subtopic 2.4 or perform a subtopic 2.5.
- In the materials and methods section, include a subsection on experimental design and statistical analysis, since it deals with the significant differences in Figure 7 of results.
- In the Results and Discussion section, figure 2, I forgot to specify what figure "F" refers to, that is, that it matches the description in the figure name.
- In the Results and Discussion section, in the name of Figure 5, it should be better described what each spectrogram and each diffractogram represents in Figure 5a and Figure 5b.
- In the discussion of FT-IR, all the interactions that occur in the composite should be described in more depth. In addition to discussing other similar work and citing.
- In the discussion of XDR and XPS the results should be discussed further and compared with the literature and cited.
- In Figure 7 (a-f) of the results, statistical analysis must be included to see significant differences.
Author Response
Dear Reviewers,
Thank you very much for the comments to our manuscript “Preparation of cellulose-grafted acrylic acid stabilized biochar-nano-scale zero-valent iron composite for Cr(VI) removal from water”( nanomaterials-3523011). We thank you reviewers for your professional, insightful, and valuable comments. Each comment or remark has been studied carefully and correction or modification has been correspondingly made. For your convenience, the comments/remarks and the itemized responses with manuscript changes (highlights in yellow) are appended below. Thank you very much for your arduous work.
Best Regards.
Yours sincerely,
Dr. Yan Zhang
On behalf of all authors
Our responses to the reviewers’ comments are listed as follows:
Response to Reviewer #1
Comments:
Dear Editor of Nanomateroials MDPI, thank you for including me as part of your reviewing team and improving the quality of submitted manuscripts. I have thoroughly reviewed the manuscript entitled" Preparation of cellulose-grafted acrylic acid stabilized biochar-nano-scale zero-valent iron composite for Cr(VI) removal from water" and consider it suitable for your journal, after the authors make the following adjustments and respond point by point to the comments.
Response:
Thank you very much for your thorough review of our manuscript entitled “Preparation of cellulose-grafted acrylic acid stabilized biochar-nano-scale zero-valent iron composite for Cr(VI) removal from water.” We sincerely appreciate the time and effort you have dedicated to providing us with your valuable feedback. We are pleased to hear that you consider our manuscript suitable for publication in Nanomaterials, pending the suggested adjustments. We will carefully address each of your comments and provide a detailed point-by-point response to ensure that all your concerns are adequately met.
Issue 1:
In the abstract section, include a sentence at the end that exemplifies the conclusion.
Response:
Thank you very much for your professional comments. In response to your comment, we have added the following sentence to the conclusion of the abstract: “In conclusion, the BC-nZVI@Cell-g-PAA composite not only demonstrated remarkable efficiency in Cr(VI) removal but also showcased its potential for practical applications in environmental remediation, as evidenced by its sustained performance over multiple reuse cycles.”
Details:
In page 1 of the revised manuscript:
A stabilized biochar (BC)–nano-scale zero-valent iron (nZVI) composite (BC-nZVI@Cell-g-PAA) was prepared using cellulose-grafted polyacrylic acid (Cell-g-PAA) as the raw material through in-situ polymerization and liquid-phase reduction methods for the remediation of Cr(VI)-contaminated water. BC-nZVI@Cell-g-PAA was characterized by XRD, FT-IR, SEM, BET, TEM, XPS. According to the batch experiments, under optimized conditions (Cr(VI) concentration of 50 mg/L, pH=3, and dosage of 2 g/L), the BC-nZVI@Cell-g-PAA composite achieved maximum Cr(VI) removal efficiency (99.69%) within 120 mins. Notably, BC, as a carrier, achieved high dispersion of nZVI through its porous structure, effectively preventing particle agglomeration and improving reaction activity. Simultaneously, the functional groups on the surface of Cell-g-PAA provide excellent protection for nZVI, significantly suppressing its oxidative deactivation. Furthermore, the composite effectively reduced Cr(VI) to insoluble Cr(III) species and stabilized them on its surface through immobilization. The synergistic effects of physical adsorption and chemical reduction greatly contributed to the removal efficiency of Cr(VI). Remarkably, the composite exhibited excellent reusability with a removal efficiency of 62.4% after five cycles, demonstrating its potential as a promising material for remediating Cr(VI)-contaminated water. In conclusion, the BC-nZVI@Cell-g-PAA composite not only demonstrated remarkable efficiency in Cr(VI) removal but also showcased its potential for practical applications in environmental remediation, as evidenced by its sustained performance over multiple reuse cycles.
Issue 2:
In the introductory section, include the following information to reinforce the definition of biochar: "Biochar, a highly porous and carbon-rich material, is obtained through the thermal decomposition of biomass under high temperatures and oxygen-limited environments. Its outstanding physicochemical characteristics, such as customizable surface area, well-defined porous structures, and a high density of oxygen-containing functional groups." Citation to: (2024). Valorization of Agave Angustifolia Bagasse Biomass from the Bacanora Industry in Sonora, Mexico as a Biochar Material: Preparation, Characterization, and Potential Application in Ibuprofen Removal. Sustainable Chemistry, 5(3), 196-214.
Response:
Thank you very much for your professional comments. I have carefully reviewed your suggestion, and I appreciate your insight. In response to your comment, I have updated the introduction to reinforce the definition of biochar, incorporating the suggested citation as follows: “Biochar, a highly porous and carbon-rich material, is obtained through the thermal decomposition of biomass under high temperatures and oxygen-limited environments. Its outstanding physicochemical characteristics, such as customizable surface area, well-defined porous structures, and a high density of oxygen-containing functional groups”,References are also cited,We have made specific corrections in the revised manuscript.
Details:
In Page 2 of the revised manuscript:
Nano-scale zero-valent iron (nZVI) is a propitious remedy for Cr(VI) polluted water based on its large surface area, great efficiency and strong reducing capability[4]. However, its practical application is constrained by inherent challenges, such as particle aggregation and susceptibility to oxidation, which impact its efficacy. To overcome these challenges[5], studies have long explored various strategies and carriers, such as bentonite, mesoporous silica, kaolin, zeolite, activated carbon, and biochar (BC), to stabilize nZVI and reduce particle aggregation[6]. Among these, BC is a readily available porous carbonaceous material, which was prepared by pyrolyzing the organic biomass under low oxygen (O) conditions. BC is distinguished by its extensive surface area, structural stability, and potent adsorption capabilities. BC, a highly porous and carbon-rich material, is obtained through the thermal decomposition of biomass under high temperatures and oxygen-limited environments. Its outstanding physicochemical characteristics, such as customizable surface area, well-defined porous structures, and a high density of oxygen-containing functional groups[7]. BC is distinguished by its extensive surface area, structural stability, and potent adsorption capabilities. Furthermore, BC surfaces are enriched with O-based functional groups, which enhance their ability to adsorb heavy metal ions[7]. In addition, the oxygen-containing functional groups abundant on the surface of BC can enhance its ability to adsorb heavy metal ions[8]. Specifically, discarded jujube branches have garnered increasing interest as BC precursors due to their porous structure and abundant cellulose content. Converting this agricultural waste into BC through pyrolysis can allow effective usage of agricultural waste products and offer a sustainable solution in order to treat the Cr-contaminated water bodies[9].
[7]Ruiz-Velducea, H.A.; Moreno-Vásquez, M.J.; Guzmán, H.; Esquer, J.; Rodríguez-Félix, F.; Graciano-Verdugo, A.Z.; Santos-Sauceda, I.; Quintero-Reyes, I.E.;Barreras-Urbina, C.G.; Vásquez-López, C.; Burruel-Ibarra, S. E.;Ozuna-Valencia,K.H.; Tapia-Hernández, J.A. Valorization of Agave angustifolia Bagasse Biomass from the Bacanora Industry in Sonora, Mexico as a Biochar Material: Preparation, Characterization, and Potential Application in Ibuprofen Removal. Sustainable Chemistry. 2024, 5,196-214.
Issue 3:
In the materials and methods section I recommend making an image that represents the design of Synthesis of "BC-nZVI@Cell-g-PAA composites", you can do it separately or in the same figure 1.
Response:
Thank you very much for your comments. We agree to create an illustration representing the design of the BC-nZVI@Cell-g-PAA composite material synthesis, and we are sorry for our unclear description. We will follow your suggestion and include a diagram in the Materials and Method section to more clearly illustrate the Synthesis Design of BC-nZVI@Cell-g-PAA Composite. In order to better illustrate the synthesis process and structural design of the composite material, we have combined this diagram with Figure 1. Additionally, the "three-neck flask" mentioned in section 2.2.1 has been changed to "beaker." Figure 1 has been moved from section 2.2.1 to section 2.2.2, and a statement has been added at the end to objectively inform the readers that the preparation flowchart of the composite material will be shown in Figure 1. The relevant modifications have been clearly presented in the revised manuscript.
Details:
In Page 4 of the revised manuscript: Figure 1
In this study, a BC-nZVI composite material with a carbon-to-iron ratio (C:Fe) of 1:1 was prepared. Specifically, 0.50 g of BC was dissolved in 2.48 g of FeSOâ‚„·7Hâ‚‚O and placed in a three-neck flask beaker. The mixture was then subjected to ultrasonic treatment to ensure thorough dispersion of the biochar. Subsequently, NaBHâ‚„ solution was slowly added dropwise to the mixture until bubbling ceased, indicating the completion of the reaction. Finally, the product was washed twice with ultrapure water to obtain the BC-nZVI sample.
2.2.2. Synthesis of BC-nZVI@Cell-g-PAA composites
50 mL aqueous solution containing 4.8 g of cellulose in a 500 mL flask was prepared under mechanical stirred at 90°C for 30 min. The above flask was equipped with a reflux condenser, stirrer, thermometer, and nitrogen line. When the mixture was quenched to 35-40°C, 100 mL n-hexane and 0.12 g of PGFE (a suspending stabilizer) was added to and agitated for 15 min. After that, potassium persulfate initiator was introduced and the solution was further stirred for 20 min and added with partially neutralized acrylic acid. Then, 2.5 mL of N,N-methylene bisacrylamide cross-linking agent (5 g/L) were added. The polymerization procession was carried out at 70°C for 3 h. The BC-nZVI powder is then mixed with Cell-g-PAA solution( the mass ratio of BC to Cell-g-PAA is 2:1). This product was stirred at ambient temperature for 30 min and then dried freeze-drying equipment at -60°C for 8 h in vacuum. The preparation flowchart of the BC-nZVI@Cell-g-PAA composite material is shown in Figure 1.
Figure 1. Schematic diagram of the preparation process of BC.
Figure 1. Schematic diagram of the preparation process of BC-nZVI@Cell-g-PAA
Issue 4:
In the materials and methods section, include a section that specifies how the absorption isotherms were carried out, it can be within subtopic 2.4 or perform a subtopic 2.5.
Response:
Thank you very much for your constructive comment. Based on your suggestion, we have created a new subsection 2.5, which provides a detailed and precise introduction to two adsorption isotherms and two adsorption kinetics models, along with the fitting equations of these models and the meanings of the parameters in the equations. Then, we removed the repetitive content from sections 3.3 and 3.4 and re-cited the references accordingly.
Details:
In Pages 5-6 of the revised manuscript:
2.5. Isotherm and adsorption kinetics
To clarify the removal mechanism of Cr(VI) by the BC-nZVI@Cell-g-PAA composite material, adsorption isotherms and adsorption kinetics models were employed to fit the experimental data. Within the scope of adsorption isotherm models, the Langmuir and Freundlich models, which are among the most widely used classical models, were applied to explain the equilibrium distribution relationship of the solute between the solid and liquid phases during the adsorption process. In terms of adsorption kinetics models, the pseudo-first-order (PFO) and pseudo-second-order (PSO) models were selected to analyze the adsorption rate changes over time, further elucidating the rate-controlling steps and the reaction mechanisms of the adsorption process.
|
(1) |
|
(2) |
|
|
The fitting equations of the two adsorption isotherm models can be expressed as follows[17]:
where (mg/L) is the equilibrium concentration of Cr(VI); (L/mg) is the equilibrium constant of the Langmuir isotherm model; (mg/g) is the equilibrium constant of the Freundlich isotherm model; and 1/n represents the adsorption intensity, such that n<1 indicates high adsorption difficulty, 1<n<2 indicates moderate adsorption difficulty, and 2<n<10 indicates high adsorption intensity and relatively easy adsorption[18].
|
(3) |
|
(4) |
The fitting equations of the two adsorption kinetics models can be expressed as follows[19]:
Here, (mg/g) signifies the adsorption capability at time t, (mg/g) is the adsorption capability at equilibrium, and (min–1) and k2 (g·mg-1·min-1) represent the rate constants for the pseudo-first-and pseudo-second-order adsorption kinetics.
[17] Zhang, H.M.; Ruan, Y.; Liang, A.P.; Shih, K.M.; Diao, Z.H.; Su, M.H.; Hou, L.A.; Chen, D.Y. Carbothermal reduction for preparing nZVI/BC to extract uranium: insight into the iron species dependent uranium adsorption behavior. J Clean Prod. 2019,239, 117873.
[18] Yang, J.W.; Tan, X.P.; Shaaban, M.; Cai, Y.J.; Wang, B.Y.; Peng, Q.A. Remediation of Cr(VI)-Contaminated Soil by Biochar-Supported Nanoscale Zero-Valent Iron and the Consequences for Indigenous Microbial Communities, Nanomaterials. 2022,12, 3541.
[19] Shi, L.N.; Lin, Y.M.; Zhang, X.; Chen, Z.L. Synthesis, characterization and kinetics of bentonite supported nZVI for the removal of Cr(VI) from aqueous solution. Chem. Eng. J. 2011,171, 612-617.
In Pages 12-13 of the revised manuscript:
3.3. Cr(VI) adsorption isotherms
The Cr(VI) adsorption capacity of BC-nZVI@Cell-g-PAA at a temperature of 25oC, BC-nZVI@Cell-g-PAA dosage of 2 g/L, and reaction time of 2 h was investigated by two different isotherm models, the Langmuir isotherm model and Freundlich adsorption isotherm model. The fitting equations for the two models can be expressed as[38] The fitting results are shown in Figure 8 and Table 4.
|
((5) |
|
((6) |
where (mg/L) is the equilibrium concentration of Cr(VI); (L/mg) is the equilibrium constant of the Langmuir isotherm model; (mg/g) is the equilibrium constant of the Freundlich isotherm model; and 1/n represents the adsorption intensity, such that n<1 indicates high adsorption difficulty, 1<n<2 indicates moderate adsorption difficulty, and 2<n<10 indicates high adsorption intensity and relatively easy adsorption[39].
3.4. Cr(VI) adsorption kinetics
|
((7) |
|
((8) |
Under a temperature of 25oC, initial Cr(VI) concentration of 50 mg/L, and BC-nZVI@Cell-g-PAA dosage of 2 g/L, the removal kinetics of Cr(VI) were analyzed using two different models ,pseudo-first-order kinetic model and pseudo-second-order kinetic models, respectively, to investigate the adsorption and reduction characteristics of BC-nZVI@Cell-g-PAA towards Cr(VI). The corresponding equations expressed as:[41] (Fig. 9 and Table 5).
Here, (mg/g) signifies the adsorption capability at time t, (mg/g) is the adsorption capability at equilibrium, and (min–1) and k2 (g·mg-1·min-1) represent the rate constants for the pseudo-first-and pseudo-second-order adsorption kinetics. The experimental data was fitted according to Eqs. (4) and (5) (Fig. 9) and the corresponding parameters are shown in Table 5. For BC-nZVI@Cell-g-PAA, the R2 values for the pseudo-first-order kinetic linear fitting for were 0.87, while that for the pseudo-second-order kinetic linear fitting were 0.99, suggesting a stronger fit for the latter. This indicated that the removal process was strongly associated with the adsorption sites on the surface of BC-nZVI@Cell-g-PAA and with chemical adsorption played a dominant role in the reaction procedure[52].
Issue 5:
In the materials and methods section, include a subsection on experimental design and statistical analysis, since it deals with the significant differences in Figure 7 of results.
Response:
Thank you very much for your advice. In Section 2.4 of the Materials and Methods, we focused on the Cr(VI) removal experiment. However, we apologize for not providing a detailed explanation of the experimental design and statistical analysis at that time. To ensure the accuracy and reliability of the research results, we have made some additions. These include a more detailed description of the Cr(VI) removal experiment, covering the precise preparation methods for each solution, a deeper analysis of the principle behind Cr(VI) concentration determination, as well as the use of error bar analysis for the single-factor experiments. These supplementary details have been added and revised in the appropriate sections of the main text.
Details:
In Page 5 of the revised manuscript:
Batch experiments were performed in 250 mL on conical flask on a reciprocating shaker at 25°C with a rotary speed of 150 rpm. The preparation of Cr(VI) solution involves dissolving K2Cr2O7 in distilled water to prepare a stock solution of 1000 mg/L Cr(VI). Prior to use, the stock solution is diluted with deionized water, and the initial pH of the solution is adjusted using 0.1 mol/L NaOH and 0.1 mol/L HNO3. Typically, 2 g/L BC-nZVI@Cell-g-PAA was added into 100 mL of 50 mg/L Cr(VI) solution. At specific time intervals, 1 mL reaction samples were obtained with the filtration, and then sent to UV spectrophotometer(Shimazu UV-2450) to exam at once. The concentration of Cr(VI) is determined using the diphenylcarbazide spectrophotometric method[16]. The principle is that diphenylcarbazide acts as a color reagent, which reacts with Cr(VI) under acidic conditions to form a purple-red compound. Under acidic conditions, Cr(VI) reacts with diphenylcarbazide to form diphenylcarbazone, while the reduced Cr(III) product forms a complex with diphenylcarbazone, producing a purple-red substance. This product has a maximum absorption at a wavelength of 540 nm, and the absorbance is linearly related to the Cr(VI) concentration within a certain range. The specific experimental procedure is as follows: (1) Filter the sample and transfer 1 mL of the supernatant into a 50 mL colorimetric tube, then dilute to the mark. (2) Add 0.5 mL of sulfuric acid and phosphoric acid (1:1), followed by 2 mL of the color reagent solution, and shake to mix. (3) After standing for 5-10 min, measure the absorbance using a 3 cm cuvette, with water as the reference, at a wavelength of 540 nm using a UV spectrophotometer. This allows the determination of the Cr(VI) concentration. Tiphenylcarbazide spectrophotometry was applied to determine the Cr(VI) concentration[16]. The Cr(VI) removal rate, R (%), was calculated from Eq.(1). All the batch experiments were repeated in triplicate, and an average was obtained. In the single-factor experimental section, we used the error bar analysis method to process the data. This approach visually presents the degree of data dispersion and uncertainty, making the analysis of experimental results more scientific and rigorous.
|
((9) |
where C0 and Ct are the initial Cr(VI) concentration and Cr(VI) concentration at time t, respectively.
Issue 6:
In the Results and Discussion section, figure 2, I forgot to specify what figure "F" refers to, that is, that it matches the description in the figure name.
Response:
Thank you for pointing this out. In Figure 2, we inadvertently omitted the label for the meaning of "F." This was an oversight on our part, and we sincerely apologize for the mistake. To avoid any confusion for the readers, we have now included an explanation of what "F" represents in the relevant section of the main text.
Details:
In Page 6 of the revised manuscript:
Figure 2. The SEM (a, b, c, d, e, f) images of BC, nZVI, BC-nZVI, Cell-g-PAA, BC-nZVI@Cell-g-PAA and BC-nZVI@Cell-g-PAA after Cr(VI) removal.
Issue 7:
In the Results and Discussion section, in the name of Figure 5, it should be better described what each spectrogram and each diffractogram represents in Figure 5a and Figure 5b.
Response:
Thank you very much for your constructive comment. We apologize for the lack of clarity in our descriptions of the spectra and diffraction patterns shown in Figures 5a and 5b. To ensure that readers can more accurately and clearly understand the information presented in these figures, we have added detailed explanations of what each spectrum and diffraction pattern represents. These revisions have been incorporated into the corresponding sections of the main text.
In Pages 9-10 of the revised manuscript:
Fig. 5a displays the FT-IR absorption spectra of BC, BC-nZVI,BC-nZVI@Cell-g-PAA and RBC-nZVI-PAA. For BC, the peaks observed in the spectrum are primarily associated with the functional groups of the biochar matrix. In the BC-nZVI spectrum, the main absorption peaks appear at 3430 cm–1 (–OH stretching vibration), 1620 cm–1 (C=C stretching vibration), 1720 cm–1 (C=O stretching vibration), and 1160 cm–1 (C–O bending vibration)[25]. As for the BC-nZVI@Cell-g-PAA composite material, the intensities of the peaks at 3430 cm–1 (–OH) and 1720 cm–1 (C=O) are significantly higher than those of BC, indicating an increase in oxygen-containing functional groups, which could enhance the adsorption of Cr(VI).The band at 3430 cm–1 can be related to -OH stretching vibration. The band at 1620 cm–1can be ascribed to C=C stretching vibration. The band at 1720 cm–1 is related to C=O stretching vibration of esters. The band at 1160 cm–1 is attributed to the flexural vibrations of C–O bonds[20]. The faint band 867 cm–1 can be ascribed to C-H flexural vibrations. The intensity of peaks corresponding to O-based functional groups, such as –OH, C=O, and C–O, in was significantly higher than that in BC. Therefore, it can be inferred that BC-nZVI@Cell-g-PAA has more O-based functional groups[26], which can accelerate the rapid adsorption of Cr(VI) ions from the polluted solution. In addition, the peak intensities of phenolic-OH and C-O for BC-nZVI@Cell-g-PAA decrease after reacting with Cr(VI)[27]. The complex formed by phenolic-OH and Cr(VI) acts as an electron shuttle, facilitating the electron transfer.
Fig.5b presents the XRD patterns of all samples (BC, nZVI, BC-nZVI, BC-nZVI@Cell-g-PAA and RBC-nZVI@Cell-g-PAA). For BC, two strong diffraction peaks emerge at 2θ=26.0o and 43.1o which are correspondence with the (002) and (100) crystal planes of the graphite structure, respectively, it shows that there is graphite component in biochar. A characteristic peak exhibited at 44.6o in nZVI, BC-nZVI, and BC-nZVI@Cell-g-PAA, indicating the presence of metallic Fe0, which was confirmed to be nZVI through comparison with Power Diffraction File (PDF) standard cards (International Center for Diffraction Data). Compared to nZVI, the peak of Fe0 in BC-nZVI became weak as BC introduced; it further decreased for BC-nZVI@Cell-g-PAA due to encapsulation of the composite by the Cell-g-PAA polymer material. Moreover, new diffraction peaks appeared in the BC-nZVI@Cell-g-PAA XRD pattern at 2θ=14.1°, 27.0°, 36.2°, and 46.7°. These peaks are associated with FeO(OH) and correspond to the (020), (120), (031), and (051) crystal planes in the PDF card. This indicates that Fe0 has undergone a redox reaction and has been converted into FeO(OH)[28].The distinct peaks of BC-nZVI@Cell-g-PAA revealed at 2θ values: are 14.1o, 27.0o, 36.2o, and 46.7o, belonging to the (020), (120), (031), and (051), respectively. These peaks, as identified using the PDF standard cards, were found to be associated with FeO(OH), indicating the subsequent occurrence of a redox reaction[23]. This observation corroborate the HRTEM and SEM results, further confirming the presence of metal Fe crystal facets in the composite.
Issue 8:
In the discussion of FT-IR, all the interactions that occur in the composite should be described in more depth. In addition to discussing other similar work and citing.
Response:
Thank you very much for your constructive comment. Based on your feedback, we conducted an in-depth exploration of the interactions occurring in the composite material and supplemented the manuscript with relevant issues, along with appropriate references. FT-IR analysis revealed that functional groups such as C=O and -OH can form stable complexes with Cr(VI), thereby enhancing the adsorption performance of the BC-nZVI@Cell-g-PAA composite material for Cr(VI). Furthermore, after the reaction with Cr(VI), the intensity of the absorption peaks for -OH and C-O in the BC-nZVI@Cell-g-PAA composite material decreased, indicating that Cr(VI) reacted with these functional groups, promoting the reduction of Cr(VI through electron transfer, which further improved the removal efficiency of Cr(VI). A detailed explanation has been added to the revised manuscript.
Details:
In Pages 9-10 of the revised manuscript:
Fig. 5a displays the FT-IR absorption spectra of BC, BC-nZVI,BC-nZVI@Cell-g-PAA and RBC-nZVI-PAA. For BC, the peaks observed in the spectrum are primarily associated with the functional groups of the biochar matrix. In the BC-nZVI spectrum, the main absorption peaks appear at 3430 cm–1 (–OH stretching vibration), 1620 cm–1 (C=C stretching vibration), 1720 cm–1 (C=O stretching vibration), and 1160 cm–1 (C–O bending vibration)[25]. As for the BC-nZVI@Cell-g-PAA composite material, the intensities of the peaks at 3430 cm–1 (–OH) and 1720 cm–1 (C=O) are significantly higher than those of BC, indicating an increase in oxygen-containing functional groups, which could enhance the adsorption of Cr(VI). Specifically, functional groups such as C=O and –OH can form stable complexes with Cr(VI), thereby improving the material's ability to adsorb Cr(VI). This is consistent with findings from previous studies [26] that showed oxygen-containing functional groups enhance the material’s hydrophilicity and iron chelation capacity. In particular, when dealing with Cr(VI)-contaminated water, the presence of oxygen-containing functional groups can accelerate the removal process of Cr(VI). The band at 3430 cm–1 can be related to -OH stretching vibration. The band at 1620 cm–1can be ascribed to C=C stretching vibration. The band at 1720 cm–1 is related to C=O stretching vibration of esters. The band at 1160 cm–1 is attributed to the flexural vibrations of C–O bonds[20]. The faint band 867 cm–1 can be ascribed to C-H flexural vibrations. The intensity of peaks corresponding to O-based functional groups, such as –OH, C=O, and C–O, in was significantly higher than that in BC. Therefore, it can be inferred that BC-nZVI@Cell-g-PAA has more O-based functional groups[27], which can accelerate the rapid adsorption of Cr(VI) ions from the polluted solution. In addition, the peak intensities of phenolic-OH and C-O for BC-nZVI@Cell-g-PAA decrease after reacting with Cr(VI)[28]. The complex formed by phenolic-OH and Cr(VI) acts as an electron shuttle, facilitating the electron transfer. This electron transfer mechanism facilitates the reduction process of Cr(VI), further enhancing the removal efficiency of Cr(VI). Similar mechanisms have been validated in other studies, particularly when oxygen-containing functional groups in materials interact with Cr(VI). These functional groups not only participate in the adsorption process but also promote the occurrence of electron transfer reactions, which is crucial for improving the overall Cr(VI) removal performance[29].
[26] Liu, P.; Wang, X.; Ma, J.; Liu, H.; Ning, P. Highly efficient immobilization of NZVI onto bio-inspired reagents functionalized polyacrylonitrile membrane for Cr(VI) reduction. Chemosphere.2019,220,1003-1013.
[29] Alsamhary, K.E. Moringa oleifera seed based green synthesis of copper nanoparticles: Characterization, environmental remediation and antimicrobial activity. Saudi J. Biol. Sci.2023,11,103820.
Issue 9:
In the discussion of XRD and XPS the results should be discussed further and compared with the literature and cited.
Response:
Thank you very much for your advice, regarding the discussion of the XRD and XPS results you mentioned, we have further expanded the analysis of these data and compared them with the results from relevant literature. Additionally, we will cite the pertinent references to ensure that the discussion is more comprehensive and well-supported. We have made the revisions in the manuscript accordingly.
Details:
In Pages 10-12 of the revised manuscript:
Fig.5b presents the XRD patterns of all samples (BC, nZVI, BC-nZVI, BC-nZVI@Cell-g-PAA and RBC-nZVI@Cell-g-PAA). For BC, two strong diffraction peaks emerge at 2θ=26.0o and 43.1o which are correspondence with the (002) and (100) crystal planes of the graphite structure, respectively, it shows that there is graphite component in biochar. This observation is consistent with previous studies, which have also identified similar diffraction peaks in biochar derived from various biomass sources[30].A characteristic peak exhibited at 44.6o in nZVI, BC-nZVI, and BC-nZVI@Cell-g-PAA , indicating the presence of metallic Fe0, which was confirmed to be nZVI through comparison with Power Diffraction File (PDF) standard cards (International Center for Diffraction Data). Compared to nZVI, the peak of Fe0 in BC-nZVI became weak as BC introduced; it further decreased for BC-nZVI@Cell-g-PAA due to encapsulation of the composite by the Cell-g-PAA polymer material. This phenomenon has also been reported in similar studies, where polymer coatings or carriers have been demonstrated to reduce the crystallinity of nZVI[31]. Moreover, new diffraction peaks appeared in the BC-nZVI@Cell-g-PAA XRD pattern at 2θ=14.1°, 27.0°, 36.2°, and 46.7°. These peaks are associated with FeO(OH) and correspond to the (020), (120), (031), and (051) crystal planes in the PDF card. This indicates that Fe0 has undergone a redox reaction and has been converted into FeO(OH)[32]. This is consistent with the findings reported in [33].The distinct peaks of BC-nZVI@Cell-g-PAA revealed at 2θ values: are 14.1o, 27.0o, 36.2o, and 46.7o, belonging to the (020), (120), (031), and (051), respectively. These peaks, as identified using the PDF standard cards, were found to be associated with FeO(OH), indicating the subsequent occurrence of a redox reaction[23]. This observation corroborate the HRTEM and SEM results, further confirming the presence of metal Fe crystal facets in the composite.
XPS study was employed to explicit the interaction between the BC-nZVI@Cell-g-PAA and Cr(VI). As shown in Fig. 6a, the characteristic peaks for O 1s, Fe 2p, and Cr 2p were detected, which confirming BC-nZVI@Cell-g-PAA to be primarily composed of C, O, and Fe. Consistent with SEM and EDS results, it was evidently that the Cr 2p was presented on the surface of BC-nZVI@Cell-g-PAA after during the removal processes[34]. Subsequently, the composition ratios and states of these elements were investigated by the narrow scans performed for O 1s, Fe 2p, and Cr 2p.
From the XPS spectrum in Figure 6(b), three peaks are observed for the O 1s signal, with peak centers at 530.1 eV, 531.7 eV, and 533.2 eV, corresponding to the characteristic peaks of oxides (Fe-O), surface hydroxyl groups (C-O/OH), and adsorbed water (H2O), respectively. After reacting with Cr(VI), FeO(OH) was formed. It can be seen from the figure that the intensities of the phenolic OH and C-O peaks in the BC-nZVI@Cell-g-PAA after reacting with Cr(VI) are weakened, particularly the phenolic OH. As reported by Chen et al.[35], after complexing with Cr(VI), the Mulliken charge of the aromatic ring significantly increases, which allows for the electrostatic interaction and adsorption of Cr(VI). This is also consistent with the previous FT-IR analysis.
As in Fig.6c and Tab. 3, peaks at 711.2 and 724.3 eV were corresponded to Fe 2p3/2 and Fe 2p1/2, respectively. What’s more, the spikes at 713.7 and 726.8 eV are indicative of Fe(III). Weak characteristic peaks at 707.3 and 720.3 eV correspond to Fe0. The peaks appeared at 718.8 and 729.5 eV correspond to satellite peaks. Before reacting with Cr(VI), the surface of BC-nZVI@Cell-g-PAA consisted of 5.1% Fe0, 78.2% Fe(II), and 16.7% Fe(III). The proportion of Fe0 reduced to 0%, Fe(II) reduced to 64.1%, and Fe(III) increased to 35.9% after the reaction. This result indicated Fe0 and Fe(II) is the primary factors to reduce the Cr(VI).This is consistent with previous studies. Literature [36] indicates that Fe0 and Fe(II) play a key role in the reduction of Cr(VI) to Cr(III) through an electron transfer mechanism. The oxidation of Fe0 and Fe(II) to Fe(III) further supports this reduction process.
As in Fig.6d, two peaks at 577.3 and 586.7 eV were attributed to Cr(III) 2p1/2 and Cr(III) 2p3/2, respectively. Moreover, the peaks at 580.0 and 589.3 eV correspond to Cr(VI) 2p3/2 and Cr(VI) 2p1/2, respectively. Among these, the proportions of Cr(III) and Cr(VI) were 85.1% and 14.9%, respectively. The simultaneous presence of Cr(III) and Cr(VI) on the surface of BC-nZVI@Cell-g-PAA after the reaction suggested that the removal processes included into both physical adsorption of Cr(VI) onto the composite surface and its chemical reduction to Cr(III). Chemical reduction is the predominant mechanism for Cr(VI) removal according to the larger proportion of Cr(III). This is in agreement with the findings of [37]. Similar studies have also emphasized the importance of chemical reduction in the removal of Cr(VI), particularly in systems involving nZVI-based composites.
[30] Wang, T.; Sun, Y.; Bai, L.; Han, C.; Sun, X. Ultrafast removal of Cr(VI) by chitosan coated biochar-supported nano
zero-valent iron aerogel from aqueous solution: Application performance and reaction mechanism. Sep. Purif.2023,306,122631.
[31] Xie, L.; Ma, Q.; Chen, Q.; Liu, Y.; Guo, P.; Zhang, J.; Duan, G.; Lin, A.; Zhang, T.; Li, S. Efficient remediation of different concentrations of Cr-contaminated soils by nano zero-valent iron modified with carboxymethyl cellulose and biochar. J. Environ. Sci.2025,147,474-486.
[33]. Gao, Y.; Yang, X.; Lu, X.; Li,M.; Wang, L.; Wang, Y. Kinetics and Mechanisms of Cr(VI) Removal by nZVI: Influencing Parameters and Modification. Catalysts. 2022,12,999.
[35] Chen, N.; Cao, S.; Zhang, L.; Peng, X.; Wang, X.; Ai, Z.; Zhang, L. Structural dependent Cr(VI) adsorption and reduction of biochar: hydrochar versus pyrochar. Sci. Total Environ. 2021, 783,147-184.
Issue 10:
In Figure 7(a-f) of the results, statistical analysis must be included to see significant differences.
Response:
Thank you very much for your professional comments. Based on your suggestion, we conducted appropriate statistical analysis of the data to assess the significance of inter-group differences, and the results were presented in the form of error bars. Accordingly, Figure 7 has been redrawn and replaced in the manuscript.
Details:
In Page 13 of the revised manuscript, Figure 7
Figure 7. Influences of (a) adsorbent dosage, (b) pH, (c) time, (d) initial concentration, (e) temperature, and (f) frequency on Cr(VI) removal by BC-nZVI@Cell-g-PAA.
Special response to Editorial Office and reviewers:
The numbers of some sections in results and discussion, figures, tables, and references have been renumbered for the reorganization of the present manuscript. We are sorry for any inconvenience caused by that. At last, we would like to thank you sincerely for your valuable suggestions and efforts that have been made during the review.

Reviewer 2 Report
Comments and Suggestions for Authors
find the attached word file

Author Response
Dear Reviewers,
Thank you very much for the comments to our manuscript “Preparation of cellulose-grafted acrylic acid stabilized biochar-nano-scale zero-valent iron composite for Cr(VI) removal from water”( nanomaterials-3523011). We thank you reviewers for your professional, insightful, and valuable comments. Each comment or remark has been studied carefully and correction or modification has been correspondingly made. For your convenience, the comments/remarks and the itemized responses with manuscript changes (highlights in yellow) are appended below. Thank you very much for your arduous work.
Best Regards.
Yours sincerely,
Dr. Yan Zhang
On behalf of all authors
Our responses to the reviewers’ comments are listed as follows:
Response to Reviewer #2
Comments:
In this manuscript Wang et al presented “Preparation of cellulose-grafted acrylic acid stabilized biochar-nano-scale zero-valent iron composite for Cr(VI) removal from water”. This work was systematically examined for hexavalent chromium removal from water based on biochar/Fe nanocomposites. However, needs developments to meet the journal quality. I recommend for major revision and my comments are:
Response:
Thank you for your constructive and fair comments. We benefited so much from them. We have carefully read the questions and comments, responded to these suggestions point-by-point, and revised the manuscript accordingly.
Issue 1:
In abstract, the hexavalent chromium environmental impact is missing.
Response:
Thank you very much for your professional comments. Regarding the issue you mentioned about the lack of discussion on the environmental impact of hexavalent chromium in the abstract, we agree with your viewpoint. In the manuscript, we have added the potential environmental impacts of Cr(VI) to ensure that the abstract is more comprehensive and covers the environmental context of the study.
Details:
In Page 1 of the revised manuscript:
Abstract: A stabilized biochar (BC)–nano-scale zero-valent iron (nZVI) composite (BC-nZVI@Cell-g-PAA) was prepared using cellulose-grafted polyacrylic acid (Cell-g-PAA) as the raw material through in-situ polymerization and liquid-phase reduction methods for the remediation of Cr(VI)-contaminated water. BC-nZVI@Cell-g-PAA was characterized by XRD, FT-IR, SEM, BET, TEM, XPS. According to the batch experiments, under optimized conditions (Cr(VI) concentration of 50 mg/L, pH=3, and dosage of 2 g/L), the BC-nZVI@Cell-g-PAA composite achieved maximum Cr(VI) removal efficiency (99.69%) within 120 min. Notably, BC, as a carrier, achieved high dispersion of nZVI through its porous structure, effectively preventing particle agglomeration and improving reaction activity. Simultaneously, the functional groups on the surface of Cell-g-PAA provide excellent protection for nZVI, significantly suppressing its oxidative deactivation. Furthermore, the composite effectively reduced Cr(VI) to insoluble Cr(III) species and stabilized them on its surface through immobilization. The synergistic effects of physical adsorption and chemical reduction greatly contributed to the removal efficiency of Cr(VI). Remarkably, the composite exhibited excellent reusability with a removal efficiency of 62.4% after five cycles, demonstrating its potential as a promising material for remediating Cr(VI)-contaminated water. In conclusion, the BC-nZVI@Cell-g-PAA composite not only demonstrated remarkable efficiency in Cr(VI) removal but also showcased its potential for practical applications in environmental remediation, as evidenced by its sustained performance over multiple reuse cycles. Moreover, Cr(VI), a toxic and carcinogenic substance, poses significant risks to aquatic ecosystems and human health, underscoring the importance of developing effective methods for its removal from contaminated water.
Issue 2:
In Introduction part, recommended to include hexavalent chromium health and environmental impact to easy understand the readers.
Response:
Thank you very much for your constructive comments. Based on your insightful advice, we have added content regarding the health and environmental impacts of Cr(VI) to help readers gain a deeper understanding of its hazards and clarify the research background. The relevant changes have been made in the manuscript.
Details:
In Page 3 of the revised manuscript:
Cr(VI) is a highly carcinogenic and mutagenic heavy metal that poses serious threats to human health and the environment. Due to its high solubility and strong oxidative properties, Cr(VI) easily spreads in water bodies, endangering aquatic life and potentially affecting human health through the food chain. Prolonged exposure can lead to skin diseases, respiratory disorders, and cancers[15]. The pollution caused by Cr(VI) not only harms ecosystems but also threatens the safety of agricultural irrigation and drinking water, necessitating the urgent development of efficient and sustainable remediation methods. The aim of this research was to investigate how Cell-g-PAA encapsulated BC-nZVI@Cell-g-PAA to improve the removal of Cr(VI) from water. Through harnessing the stability of BC and the protective attributes of Cell-g-PAA, the issue of nZVI particle aggregation could be tackled in order to enhanced dispersal and stability. The outcomes of this study hold significant importance in formulating effective strategies for Cr pollution control in aquatic environments.
Issue 3:
In page no.3, fig.1 second step, the KOH activation process for BC preparation needs clear information’s and why this process are required.
Response:
Thanks for the comment. We apologize for the lack of clarity in the description of the KOH activation process in Step 2 of Figure 1, which may have caused confusion for readers. We have now added a more detailed explanation of the KOH activation process in the manuscript, and Figure 1 has been updated to visually highlight the activation process, making it easier for readers to clearly understand its crucial role in enhancing the material's performance.
Details:
In Pages 3-4 of the revised manuscript: Figure 1
The preparation of BC was conducted through high-temperature carbonization of the powdered jujube branch (Fig. 1). The branch powder, sourced from the Jiaxian area in Yulin, China, was subjected to low-temperature pyrolysis and high-temperature activation in a laboratory-scale tube furnace(XL-RL, Yangzhou Xingliu Electric Appliance Co., LTD). The process entailed heating the powder at 5°C/min accompanied with nitrogen atmosphere, maintained at 600oC for 2 h to produce BC. Then, obtained BC was confused with KOH (the mass proportion of BC:KOH is 1:2) in a crucible and subjected to 800oC in the tube furnace for 2 h, and then, naturally cooled and stirred in a 1 mol/L HCl solution for 4 h. Finallly, they were washed till reached a neutral pH and desiccated at 105oC. The obtained BC is mixed with KOH (with a mass ratio of BC:KOH=1:2) in a crucible, and then heated at 800°C for 2 hours in a tube furnace to obtain KOH-activated BC. KOH activation increases the porosity, surface functional groups, and specific surface area of the biochar, thereby enhancing its adsorption capacity and ability to remove pollutants. After activation, the mixture is naturally cooled and stirred in a 1 mol/L HCl solution for 4 hours. Finally, it is washed until the pH becomes neutral and dried at 105°C.
Figure 1. Schematic diagram of the preparation process of BC.
Figure 1. Schematic diagram of the preparation process of BC-nZVI@Cell-g-PAA
Issue 4:
In recent work, there are numerous research works reported based on natural substance based activated carbon preparation and used for different catalytic applications. So, what is the novelty of the developed work?
Response:
Thank you for your insightful comment. In response to your query regarding the novelty of our work, we have expanded the manuscript to clarify this aspect. This study utilizes discarded jujube tree branches from the Yulin region as a precursor for biochar. In this area, the density of jujube tree planting is high, but the discarded branches are often burned or landfilled, leading to ecological risks. By converting these branches into functional materials, we not only effectively utilize regional waste biomass but also take advantage of the unique properties of jujube tree branches. Unlike common raw materials like straw and fruit shells[1], jujube branches have distinct components, with significant differences in lignin and cellulose content and structure. This distinction is expected to impart superior performance to the biochar[2], thus offering a novel and environmentally beneficial approach to biochar preparation.
[1] Srocke, F.; Han, L.; Dutilleul, P.; Xiao, X.; Smith, D. Synchrotron X-ray microtomography and multifractal analysis for the characterization of pore structure and distribution in softwood pellet biochar. Biochar.2021,3,671-686.
[2] Liu, Q.; Song, Z.; Li, J.; Pan, C. Efficacy of Agricultural Residue-Derived Biochar for Tackling Cadmium Contamination in an Aqueous Solution. Molecules,2024,29,3545.
Details:
In Page 2 of the revised manuscript:
Nano-scale zero-valent iron (nZVI) is a propitious remedy for Cr(VI) polluted water based on its large surface area, great efficiency and strong reducing capability[4]. However, its practical application is constrained by inherent challenges, such as particle aggregation and susceptibility to oxidation, which impact its efficacy. To overcome these challenges[5], studies have long explored various strategies and carriers, such as bentonite, mesoporous silica, kaolin, zeolite, activated carbon, and biochar (BC), to stabilize nZVI and reduce particle aggregation[6]. Among these, BC is a readily available porous carbonaceous material, which was prepared by pyrolyzing the organic biomass under low oxygen (O) conditions. BC is distinguished by its extensive surface area, structural stability, and potent adsorption capabilities. biochar, a highly porous and carbon-rich material, is obtained through the thermal decomposition of biomass under high temperatures and oxygen-limited environments. Its outstanding physicochemical characteristics, such as customizable surface area, well-defined porous structures, and a high density of oxygen-containing functional groups[7]. BC is distinguished by its extensive surface area, structural stability, and potent adsorption capabilities. Furthermore, BC surfaces are enriched with O-based functional groups, which enhance their ability to adsorb heavy metal ions[7]. In addition, the oxygen-containing functional groups abundant on the surface of BC can enhance its ability to adsorb heavy metal ions[8]. The jujube tree planting density in the Yulin region is high, yet the discarded branches are typically burned or landfilled, which poses ecological risks. By converting these branches into functional materials, the regional waste biomass can be effectively utilized. Therefore, biochar (BC), due to its porous structure and rich cellulose content, has gained increasing interest as a precursor from discarded jujube branches. Specifically, discarded jujube branches have garnered increasing interest as BC precursors due to their porous structure and abundant cellulose content. Converting this agricultural waste into BC through pyrolysis can allow effective usage of agricultural waste products and offer a sustainable solution in order to treat the Cr-contaminated water bodies[9].
Issue 5:
The activated carbon preparation, Jujube twing used as a carbon source, is there any specific reason?
Response:
Thank you for your thoughtful question. The use of jujube twigs as a carbon source in our study is based on both environmental and material-specific considerations. Jujube tree branches are abundant in the Yulin region, yet they are often discarded through burning or landfilling, which poses ecological risks. By utilizing these twigs as a precursor for activated carbon, we not only address the issue of regional waste biomass but also take advantage of their unique composition. Compared to more commonly used carbon sources, such as straw or fruit shells, jujube twigs have distinct lignin and cellulose content, which we believe may enhance the structural and functional properties of the resulting activated carbon, similar to the previous question.
Issue 6:
In fig.2 a SEM images, what is the average surface area and pore size?
Response:
Thank you for your valuable comment. Regarding your question, SEM images typically provide morphological information of materials, including surface structure and pore morphology. However, they cannot directly measure the average surface area and pore size of the material. To accurately obtain these parameters, additional characterization methods such as BET surface area testing and pore size distribution analysis are required. The specific analysis and characterization of BET contrast surface area and aperture have been modified and supplemented in the original manuscript.
Details:
In Pages 12-13 of the revised manuscript: Figure 7 and Table 3
Figure 7 presents the Nâ‚‚ adsorption-desorption isotherms and pore size distribution curves for four materials. As shown in Figure 7(a), the desorption curve of the original BC is nearly linear, indicating a limited pore structure. The adsorption-desorption curve of KOH-activated BC follows a Type I isotherm, with a rapid increase in adsorption at low relative pressures, suggesting the presence of a rich microporous structure[38]. However, in the relative pressure range of P/Pâ‚€=0.6-0.9, the increase in adsorption becomes less pronounced, indicating the presence of a small amount of mesoporous structures. BC-nZVI also follows a Type I isotherm. Compared to the original BC, BC-nZVI exhibits a larger adsorption volume, although still smaller than that of KOH-activated BC. This is likely due to the introduction of nZVI, which occupies part of the pore space, thus limiting the further expansion of the pore structure. BC-nZVI@Cell-g-PAA also adheres to a Type I isotherm, but its adsorption volume is smaller than that of BC-nZVI. This may be due to the modification by Cell-g-PAA, which restricts the openness of the pore structure and results in a lower adsorption capacity compared to both BC-nZVI and KOH-activated BC.
Table 3 shows the pore structure parameters of the material. The specific surface area of BC is 1987.6 m²/g, after the loading of nZVI, the specific surface area of BC-nZVI decreased to 735.6 m²/g. This reduction is likely due to the deposition of nZVI particles on the surface of BC during the liquid-phase reduction process, filling or adhering to the pore structures of the BC, which significantly reduces the material's specific surface area. Furthermore, partial oxidation of nZVI may occur during subsequent storage or usage, forming iron oxides (such as Feâ‚‚O₃ and Fe₃Oâ‚„). These oxides further adhere to the pore surfaces or inside the pore channels of BC, narrowing the pores or even completely blocking them, thereby further decreasing the specific surface area. Furthermore, after the addition of the coating layer, the specific surface area of the BC-nZVI@Cell-g-PAA composite material decreased further to 142.3 m²/g. This reduction is likely due to the introduction of Cell-g-PAA, which altered the surface morphology of BC-nZV[39]. The originally rough and porous surface was covered by a denser polymer layer, which might have infiltrated the internal pores of BC-nZVI, reducing the pore volume and thereby further decreasing the specific surface area.
Figure 7. (a) N2 adsorption-desorption isotherm diagram; (b) aperture profile.
Table 3. The specific surface area and pore structure parameters of Original BC, BC, BC-nZVI and BC-nZVI@Cell-g-PAA composites.
Sample |
SSA (m2 g–1) |
Pore volume (m3 g–1) |
|
Vtotal |
Vmic |
||
Original BC |
220.87 |
0.1 |
0.08 |
BC |
1987.6 |
0.86 |
0.66 |
BC-nZVI |
735.6 |
0.39 |
0.25 |
BC-nZVI@Cell-g-PAA |
142.3 |
0.07 |
0.12 |
[38] Cheng, S.; Wang, X.; Du, K.; Mao, Y.; Han, Y.; Li, L.; Liu, X.; Wen, G. Hierarchical Lotus-Seedpod-Derived Porous Activated Carbon Encapsulated with NiCo2S4 for a High-Performance All-Solid-State Asymmetric Supercapacitor. Molecules, 2023,28,1420-3049.
Issue 7:
In the abstract, authors described that the materials were characterized and confirmed with BET method. However, In the entire manuscript BET information are missing.
Response:
Thank you very much for your valuable suggestions. In the abstract of the paper, we mentioned the use of the BET method for material characterization. Table 3 of the full text also lists the surface area and pore size distribution data for various materials. However, due to an oversight on our part, the expression was not sufficiently clear, and we failed to include the relevant graphs for surface area and pore size distribution, which could have more intuitively presented the key information. We sincerely apologize for this. As a result, we have now added the relevant BET surface area and pore size distribution graphs, as well as the BET data for BC after KOH activation, and have made the corresponding revisions in the manuscript.
Details:
In Pages12-13 of the revised manuscript: Figure 7 and Table 3
Figure 7 presents the Nâ‚‚ adsorption-desorption isotherms and pore size distribution curves for four materials. As shown in Figure 7(a), the desorption curve of the original BC is nearly linear, indicating a limited pore structure. The adsorption-desorption curve of KOH-activated BC follows a Type I isotherm, with a rapid increase in adsorption at low relative pressures, suggesting the presence of a rich microporous structure[38]. However, in the relative pressure range of P/Pâ‚€=0.6-0.9, the increase in adsorption becomes less pronounced, indicating the presence of a small amount of mesoporous structures. BC-nZVI also follows a Type I isotherm. Compared to the original BC, BC-nZVI exhibits a larger adsorption volume, although still smaller than that of KOH-activated BC. This is likely due to the introduction of nZVI, which occupies part of the pore space, thus limiting the further expansion of the pore structure. BC-nZVI@Cell-g-PAA also adheres to a Type I isotherm, but its adsorption volume is smaller than that of BC-nZVI. This may be due to the modification by Cell-g-PAA, which restricts the openness of the pore structure and results in a lower adsorption capacity compared to both BC-nZVI and KOH-activated BC.
Table 3 shows the pore structure parameters of the material. The specific surface area of BC is 1987.6 m²/g, after the loading of nZVI, the specific surface area of BC-nZVI decreased to 735.6 m²/g. This reduction is likely due to the deposition of nZVI particles on the surface of BC during the liquid-phase reduction process, filling or adhering to the pore structures of the BC, which significantly reduces the material's specific surface area. Furthermore, partial oxidation of nZVI may occur during subsequent storage or usage, forming iron oxides (such as Feâ‚‚O₃ and Fe₃Oâ‚„). These oxides further adhere to the pore surfaces or inside the pore channels of BC, narrowing the pores or even completely blocking them, thereby further decreasing the specific surface area. Furthermore, after the addition of the coating layer, the specific surface area of the BC-nZVI@Cell-g-PAA composite material decreased further to 142.3 m²/g. This reduction is likely due to the introduction of Cell-g-PAA, which altered the surface morphology of BC-nZV[39]. The originally rough and porous surface was covered by a denser polymer layer, which might have infiltrated the internal pores of BC-nZVI, reducing the pore volume and thereby further decreasing the specific surface area.
Figure 7. (a) N2 adsorption-desorption isotherm diagram; (b) aperture profile.
Table 3. The specific surface area and pore structure parameters of Original BC, BC, BC-nZVI and BC-nZVI@Cell-g-PAA composites.
Sample |
SSA (m2 g–1) |
Pore volume (m3 g–1) |
|
Vtotal |
Vmic |
||
Original BC |
220.87 |
0.1 |
0.08 |
BC |
1987.6 |
0.86 |
0.66 |
BC-nZVI |
735.6 |
0.39 |
0.25 |
BC-nZVI@Cell-g-PAA |
142.3 |
0.07 |
0.12 |
[38] Cheng, S.; Wang, X.; Du, K.; Mao, Y.; Han, Y.; Li, L.; Liu, X.; Wen, G. Hierarchical Lotus-Seedpod-Derived Porous Activated Carbon Encapsulated with NiCo2S4 for a High-Performance All-Solid-State Asymmetric Supercapacitor. Molecules, 2023,28,1420-3049.
Issue 8:
Recommended to characterize the BET measurements each step modification and calculate the pore size and pore volume to easy understand the readers.
Response:
Thank you very much for your professional comments. Based on your suggestion, we have added the BET data for the original biochar to the manuscript, improved the information regarding the average pore diameter of the four materials, and included the adsorption-desorption curves and pore size distribution plots for the four materials. This is aimed at helping readers to better and more intuitively understand the relevant content. These additions and corresponding adjustments have now been completed in the manuscript.
Details:
In Pages12-13 of the revised manuscript: Figure 7 and Table 3
Figure 7 presents the Nâ‚‚ adsorption-desorption isotherms and pore size distribution curves of four different materials. As shown in Figure 7(a), the desorption curve of the original BC is nearly a straight line, indicating a limited pore structure. The adsorption-desorption curve of KOH-activated BC conforms to a Type I isotherm, with a rapid increase in adsorption at low relative pressure and an average pore size of 1.72 nm, suggesting a well-developed microporous structure[38]. However, in the relative pressure range of P/Pâ‚€=0.6–0.9, the increase in adsorption slows down, indicating the presence of a small amount of mesopores. Similarly, BC-nZVI also follows a Type I isotherm, with an average pore size of 2.23 nm. Compared with the original BC, BC-nZVI exhibits a larger adsorption volume, although it remains smaller than that of KOH-activated BC. This could be attributed to the introduction of nZVI, which occupies part of the pores and thus restricts further expansion of the pore structure. BC-nZVI@Cell-g-PAA also follows a Type I isotherm, with an average pore size of 2.38 nm. However, its adsorption volume is smaller than that of BC-nZVI, possibly due to the modification by Cell-g-PAA, which limits the openness of the pore structure and results in a lower adsorption capacity compared to BC-nZVI and KOH-activated BC.
Table 3 shows the pore structure parameters of the material. The specific surface area of BC is 1987.6 m²/g, after the loading of nZVI, the specific surface area of BC-nZVI decreased to 735.6 m²/g. This reduction is likely due to the deposition of nZVI particles on the surface of BC during the liquid-phase reduction process, filling or adhering to the pore structures of the BC, which significantly reduces the material's specific surface area. Furthermore, partial oxidation of nZVI may occur during subsequent storage or usage, forming iron oxides (such as Feâ‚‚O₃ and Fe₃Oâ‚„). These oxides further adhere to the pore surfaces or inside the pore channels of BC, narrowing the pores or even completely blocking them, thereby further decreasing the specific surface area. Furthermore, after the addition of the coating layer, the specific surface area of the BC-nZVI@Cell-g-PAA composite material decreased further to 142.3 m²/g. This reduction is likely due to the introduction of Cell-g-PAA, which altered the surface morphology of BC-nZV[39]. The originally rough and porous surface was covered by a denser polymer layer, which might have infiltrated the internal pores of BC-nZVI, reducing the pore volume and thereby further decreasing the specific surface area.
Figure 7. (a) N2 adsorption-desorption isotherm diagram; (b) aperture profile.
Table 3. The specific surface area and pore structure parameters of Original BC, BC, BC-nZVI and BC-nZVI@Cell-g-PAA composites.
Sample |
SSA (m2 g–1) |
Pore volume (m3 g–1) |
average pore size(nm) |
|
Vtotal |
Vmic |
|||
Original BC |
220.87 |
0.1 |
0.08 |
1.79 |
BC |
1987.6 |
0.86 |
0.66 |
1.72 |
BC-nZVI |
735.6 |
0.39 |
0.25 |
2.23 |
BC-nZVI@Cell-g-PAA |
142.3 |
0.07 |
0.12 |
2.38 |
[38] Cheng, S.; Wang, X.; Du, K.; Mao, Y.; Han, Y.; Li, L.; Liu, X.; Wen, G. Hierarchical Lotus-Seedpod-Derived Porous Activated Carbon Encapsulated with NiCo2S4 for a High-Performance All-Solid-State Asymmetric Supercapacitor. Molecules, 2023,28,1420-3049.
Issue 9:
Fig.4 HRTEM images, d(ii), d(iii) scale range are missing.
Response:
Thank you very much for your professional comments. We apologize for the missing scale ranges for d(ii) and d(iii) in the HRTEM images of Figure 4 that you mentioned. In the revised manuscript, we have added the scale ranges for these two images to better present the experimental results.
Details:
In Page 9 of the revised manuscript: Figure 4
Figure 4. (a) HRTEM thickness analysis of the Cell-g-PAA coating layer of the composite; (b) Fe particle size distribution; (c) Fe particle distribution on the composite surface; and (d)(i) HRTEM images, (ii) lattice spacing, and (iii) electron interval diffraction patterns of Fe in the composite.
Issue 10:
Fig.5, XRD results x and y axis should double check.
Response:
Thank you very much for your professional comments. In response to the issue you pointed out, we have re-verified the relevant data. The necessary revisions have been made in the manuscript to ensure the results are accurate and clearly presented. We sincerely appreciate your assistance once again.
Details:
In Page 10 of the revised manuscript: Figure 5
Figure 5. (a) FT-IR and (b)XRD patterns of different samples.
Issue 11:
In XPS results, fig.6(b) O2 spectra needs detailed explanation (it seems chemisorbed O2).
Response:
Thank you very much for your professional comments. We apologize for the oversight in missing the analysis of the O 1s spectrum. In response to your suggestion, we have now included a detailed analysis of the O 1s spectrum in the revised manuscript to present our experimental results more clearly.
Details:
In Page 11 of the revised manuscript:
From the XPS spectrum in Figure 6(b), three peaks are observed for the O 1s signal, with peak centers at 530.1 eV, 531.7 eV, and 533.2 eV, corresponding to the characteristic peaks of oxides (Fe-O), surface hydroxyl groups (C-O/OH), and adsorbed water (H2O), respectively. After reacting with Cr(VI), FeO(OH) was formed. It can be seen from the figure that the intensities of the phenolic OH and C-O peaks in the BC-nZVI@Cell-g-PAA after reacting with Cr(VI) are weakened, particularly the phenolic OH. As reported by Chen et al[35], after complexing with Cr(VI), the Mulliken charge of the aromatic ring significantly increases, which allows for the electrostatic interaction and adsorption of Cr(VI). This is also consistent with the previous FT-IR analysis.
Issue 12:
In fig.7 (a) x axis are missing, and the entire results analytical data such as error bars are missing.
Response:
Thank you very much for your professional comments. We sincerely apologize for the missing x-axis label and the omission of error bars in Figure 7(a). The necessary corrections have been made in the revised manuscript, where we have added the x-axis label, included all required error bars, and supplemented the corresponding data analysis results.
Details:
In Page 14 of the revised manuscript: Figure 8
Figure 8. Influences of (a) adsorbent dosage, (b) pH, (c) time, (d) initial concentration, (e) temperature, and (f) frequency on Cr(VI) removal by BC-nZVI@Cell-g-PAA.
Issue 13:
Specificity of the materials towards Cr(VI) reduction missing.
Response:
Thank you very much for your professional comments. We acknowledge that the specificity of Cr(VI) reduction was not adequately discussed in the original manuscript, which was an oversight on our part. We have supplemented the relevant content and made corresponding revisions to the manuscript.
Details:
In Pages 18-19 of the revised manuscript:
The BC-nZVI@Cell-PAA material, with its efficient, stable, and environmentally friendly properties, provides scientific evidence and technical support for the practical application of nano-zero-valent iron (nZVI) in the removal of Cr(VI). It demonstrates broad application potential. The pseudo-second-order kinetic model indicated that the removal process was strongly associated with the adsorption sites on the surface of BC-nZVI@Cell-g-PAA and with chemical adsorption played a dominant role in the reaction procedure. Additionally, the Langmuir model showed the monolayer adsorption of the BC-nZVI@Cell-g-PAA composite towards (Cr(VI)) happened onto a surface containing a finite number of identical sites. Most important, the absorbed BC-nZVI@Cell-g-PAA composite can be regenerated and recycled. Furthermore, the reduction of Cr(VI) demonstrates significant specificity. In terms of environmental conditions, the reduction process demands specific parameters, with the reduction rate of Cr(VI) varying across different pH levels. Typically, reduction to Cr(III) occurs more readily under acidic conditions, whereas the reaction may be inhibited under alkaline conditions. Regarding the impact of the products, Cr(VI) is highly toxic and mobile, while the reduced Cr(III) is less toxic and tends to precipitate, leading to a substantial decrease in environmental mobility and ecological toxicity. This transformation from high toxicity to low toxicity highlights the specific role of Cr(VI) reduction in areas such as environmental remediation. These results demonstrate that the unique BC-nZVI@Cell-g-PAA exhibits great potential for application in water treatment.
Special response to Editorial Office and reviewers:
The numbers of some sections in results and discussion, figures, tables, and references have been renumbered for the reorganization of the present manuscript. We are sorry for any inconvenience caused by that. At last, we would like to thank you sincerely for your valuable suggestions and efforts that have been made during the review.

Reviewer 3 Report
Comments and Suggestions for Authors
1. Numerical values of the important results needed to incorporate in the abstract as well as in conclusion.
2. Jujubi plant name sgould be highlighted in title , keywords as well as in abstract.
3. Introduction part needs to modify with incorporation of KOH importance along with the literature.
4. What is the rational behind the choosing BC:KOH ratio 1:2 ?
5. Also it would be interesting to conduct the study of the effect of Fe loading over Chromium removal.
6. In figure 1 schematic, please add the pyrolysis and activation temperature.
7. Does all the waste water bodies pH is acidic ?
8. Abbreviations should be ellaborated prior to use it first time in the manuscript.
9.Please make sure to check typo erroe through out the manuscript.
10. N2 adsorption desoprtion and particle size distribution plots should be incorporated in the manuscript with respective section.
11. Can Cell-g-PAA coating layer thickness ahave any effect over the chromium removal performance of the catalyst ?
Author Response
Dear Reviewers,
Thank you very much for the comments to our manuscript “Preparation of cellulose-grafted acrylic acid stabilized biochar-nano-scale zero-valent iron composite for Cr(VI) removal from water”( nanomaterials-3523011). We thank you reviewers for your professional, insightful, and valuable comments. Each comment or remark has been studied carefully and correction or modification has been correspondingly made. For your convenience, the comments/remarks and the itemized responses with manuscript changes (highlights in yellow) are appended below. Thank you very much for your arduous work.
Best Regards.
Yours sincerely,
Dr. Yan Zhang
On behalf of all authors
Our responses to the reviewers’ comments are listed as follows:
Response to Reviewer #3
Issue 1:
Numerical values of the important results needed to incorporate in the abstract as well as in conclusion.
Response:
Thanks for the comment. In the abstract section, we have incorporated the specific numerical values of the key results. Following your suggestion, we have also added these critical numerical data in the conclusion section. The corresponding revisions have now been completed in the manuscript.
Details:
In Page 18 of the revised manuscript:
BC-nZVI@Cell-g-PAA composites using an in-situ polymerization and liquid-phase reduction method were prepared. The BC-nZVI@Cell-g-PAA composite material was synthesized using in situ polymerization and liquid-phase reduction methods. Through optimization, it was determined that under conditions of a Cr(VI) concentration of 50 mg/L, pH of 3, and a dosage of 2 g/L, the BC-nZVI@Cell-g-PAA composite achieved the highest Cr(VI) removal efficiency of 99.69% within 120 mins. This composite material exhibits several key characteristics: One hand, BC provides a large surface area, nZVI particles are uniformly distributed on the BC surface. On another hand, Cell-g-PAA offers excellent protection, enhancing the dispersion and stability of the composite. Furthermore, the principal mechanism for eliminating Cr(VI) using BC-nZVI@Cell-g-PAA entailed the (i) conversion of Fe0 to Fe(II) and Fe(III), (ii) reduction of Cr(VI) to Cr(III), and (iii)Cr removal via adsorption, reduction, and co-precipitation. The reaction process was involved the synergistic action of physical adsorption and chemical reduction of Cr(VI) based on a pseudo-second-order kinetic model. Finally, the inclusion of a Cell-g-PAA polymer layer on the surface of nZVI increased the number of O-based functional groups, further enhancing the efficacy of Cr(VI) removal.
Issue 2:
Jujube plant name should be highlighted in title, keywords as well as in abstract.
Response:
Thanks for the comment. The manuscript has been revised according to your suggestions, with the following specific modifications: (1) The research subject "jujube branch" has been explicitly incorporated into the paper title. (2) "Jujube branch" has been added as a core keyword in the keyword section. (3) The abstract has been revised to emphasize the preparation of biochar using jujube branches as the raw material.
Details:
In Pages 1 and 2 of the revised manuscript:
Preparation of cellulose-grafted acrylic acid stabilized jujube branch biochar-supported nano-scale zero-valent iron composite for Cr(VI) removal from water
Keywords: Jujube branch; Biochar; nZVI; Cell-g-PAA; Removal; Cr(VI)
Nano-scale zero-valent iron (nZVI) is a propitious remedy for Cr(VI) polluted water based on its large surface area, great efficiency and strong reducing capability[4]. However, its practical application is constrained by inherent challenges, such as particle aggregation and susceptibility to oxidation, which impact its efficacy. To overcome these challenges[5], studies have long explored various strategies and carriers, such as bentonite, mesoporous silica, kaolin, zeolite, activated carbon, and biochar (BC), to stabilize nZVI and reduce particle aggregation[6]. Among these, BC is a readily available porous carbonaceous material, which was prepared by pyrolyzing the organic biomass under low oxygen (O) conditions. BC is distinguished by its extensive surface area, structural stability, and potent adsorption capabilities. biochar, a highly porous and carbon-rich material, is obtained through the thermal decomposition of biomass under high temperatures and oxygen-limited environments. Its outstanding physicochemical characteristics, such as customizable surface area, well-defined porous structures, and a high density of oxygen-containing functional groups[7]. BC is distinguished by its extensive surface area, structural stability, and potent adsorption capabilities. Furthermore, BC surfaces are enriched with O-based functional groups, which enhance their ability to adsorb heavy metal ions[7]. In addition, the oxygen-containing functional groups abundant on the surface of BC can enhance its ability to adsorb heavy metal ions[8]. The jujube tree planting density in the Yulin region is high, yet the discarded jujube branches are typically burned or landfilled, which poses ecological risks. By converting these branches into functional materials, the regional waste biomass can be effectively utilized. Therefore, biochar (BC), due to its porous structure and rich cellulose content, has gained increasing interest as a precursor from discarded jujube branches. Specifically, discarded jujube branches have garnered increasing interest as BC precursors due to their porous structure and abundant cellulose content. Converting this agricultural waste into BC through pyrolysis can allow effective usage of agricultural waste products and offer a sustainable solution in order to treat the Cr-contaminated water bodies[9].
Issue 3:
Introduction part needs to modify with incorporation of KOH importance along with the literature.
Response:
Thank you very much for your professional comments. In response to your suggestion, we have revised the Introduction section to incorporate the importance of KOH activation and relevant literature.
Details:
In Page 2of the revised manuscript:
Nano-scale zero-valent iron (nZVI) is a propitious remedy for Cr(VI) polluted water based on its large surface area, great efficiency and strong reducing capability[4]. However, its practical application is constrained by inherent challenges, such as particle aggregation and susceptibility to oxidation, which impact its efficacy. To overcome these challenges[5], studies have long explored various strategies and carriers, such as bentonite, mesoporous silica, kaolin, zeolite, activated carbon, and biochar (BC), to stabilize nZVI and reduce particle aggregation[6]. Among these, BC is a readily available porous carbonaceous material, which was prepared by pyrolyzing the organic biomass under low oxygen (O) conditions. BC is distinguished by its extensive surface area, structural stability, and potent adsorption capabilities. biochar, a highly porous and carbon-rich material, is obtained through the thermal decomposition of biomass under high temperatures and oxygen-limited environments. Its outstanding physicochemical characteristics, such as customizable surface area, well-defined porous structures, and a high density of oxygen-containing functional groups[7]. BC is distinguished by its extensive surface area, structural stability, and potent adsorption capabilities. Furthermore, BC surfaces are enriched with O-based functional groups, which enhance their ability to adsorb heavy metal ions[7]. In addition, the oxygen-containing functional groups abundant on the surface of BC can enhance its ability to adsorb heavy metal ions[8]. The jujube tree planting density in the Yulin region is high, yet the discarded jujube branches are typically burned or landfilled, which poses ecological risks. By converting these branches into functional materials, the regional waste biomass can be effectively utilized. Therefore, biochar (BC), due to its porous structure and rich cellulose content, has gained increasing interest as a precursor from discarded jujube branches. Specifically, discarded jujube branches have garnered increasing interest as BC precursors due to their porous structure and abundant cellulose content. Converting this agricultural waste into BC through pyrolysis can allow effective usage of agricultural waste products and offer a sustainable solution in order to treat the Cr-contaminated water bodies[9]. However, the limited specific surface area, underdeveloped pore structure, and simplistic surface chemistry of pristine biochar constrain its performance in practical applications. To address these limitations, chemical activation has become a key strategy for enhancing biochar properties. Among various activation methods, KOH activation is widely utilized in biochar modification research due to its efficiency and practicality. KOH activation not only significantly increases the specific surface area and porosity of biochar, optimizing its physical structure, but also introduces abundant oxygen-containing functional groups, thereby improving its surface chemistry[10]. This leads to a substantial enhancement in its adsorption capacity for heavy metal ions.
Issue 4:
What is the rational behind the choosing BC:KOH ratio 1:2?
Response:
Thank you very much for your professional comments. The choice of a 1:2 ratio is based on preliminary experimental results. During the pre-experimental phase, we tested different ratios, and the results showed that when the ratio was 1:2, the material exhibited the largest specific surface area and the best performance. This indicates that the 1:2 ratio achieves the optimal balance between promoting effective reactions and achieving high yield efficiency.
Details:
In Page 4 of the revised manuscript:
The preparation of BC was conducted through high-temperature carbonization of the powdered jujube branch (Fig. 1). The branch powder, sourced from the Jiaxian area in Yulin, China, was subjected to low-temperature pyrolysis and high-temperature activation in a laboratory-scale tube furnace(XL-RL, Yangzhou Xingliu Electric Appliance Co., LTD). The process entailed heating the powder at 5°C/min accompanied with nitrogen atmosphere, maintained at 600oC for 2 h to produce BC. Then, obtained BC was confused with KOH (the mass proportion of BC:KOH is 1:2) in a crucible and subjected to 800oC in the tube furnace for 2 h, and then, naturally cooled and stirred in a 1 mol/L HCl solution for 4 h. Finallly, they were washed till reached a neutral pH and desiccated at 105oC. The obtained BC is mixed with KOH (with a mass ratio of BC:KOH=1:2),a ratio determined after comparing multiple sets of preliminary experiments,and then heated at 800°C for 2 h in a tube furnace to obtain KOH-activated BC. KOH activation increases the porosity, surface functional groups, and specific surface area of the biochar, thereby enhancing its adsorption capacity and ability to remove pollutants. After activation, the mixture is naturally cooled and stirred in a 1 mol/L HCl solution for 4 h. Finally, it is washed until the pH becomes neutral and dried at 105°C.
Issue 5:
Also it would be interesting to conduct the study of the effect of Fe loading over Chromium removal.
Response:
Thank you very much for your valuable suggestion regarding the impact of iron loading on chromium removal performance. Although this aspect has not been addressed in the current study, we fully recognize its significant importance. Moving forward, we plan to make the impact of iron loading on material performance a key focus in our future research and will explore the correlations and mechanisms in depth.
Issue 6:
In figure 1 schematic, please add the pyrolysis and activation temperature.
Response:
Thank you very much for your professional comments. In accordance with your suggestion, we have redrawn Figure 1 and added labels for the pyrolysis and activation temperatures to provide more complete and clearer information. The revisions have been made in the manuscript.
Details:
In Page 5 of the revised manuscript:
Figure 1. Schematic diagram of the preparation process of BC.
Figure 1. Schematic diagram of the preparation process of BC-nZVI@Cell-g-PAA
Issue 7:
Does all the waste water bodies pH is acidic?
Response:
Thank you very much for your professional comments. Regarding the question of whether wastewater pH is generally acidic, it is important to note that not all wastewater has an acidic pH. The pH value of wastewater is influenced by multiple factors, including the source of the wastewater, the treatment processes, and the chemical composition present. In our study, to comprehensively simulate wastewater treatment under different pH conditions, we conducted experiments on the removal of Cr(VI) using composite materials in wastewater with varying pH values. The results indicated that the removal efficiency was most ideal under acidic conditions. However, it is important to emphasize that in real-world environments, wastewater pH can vary and is not necessarily always acidic.
Details:
In Page 14 of the revised manuscript: Figure 8
To more accurately simulate the actual pH environment of wastewater, we conducted experimental studies on the removal of Cr(VI) using composite materials under different pH conditions. Seen from the Fig. 8(b), the Cr(VI) removal rate of BC-nZVI@Cell-g-PAA gradually decreased with increased pH, indicating that acidic conditions are more conducive for Cr(VI) elimination than those of the basic condition. Many studies have shown that Cr(VI) of different ionic forms can be found in water solutions, including CrO42–, Cr2O72–, Cr3O102–, Cr4O132–, and HCrO4–[42]. Under acidic circumstances, the primary state of Cr(VI) is HCrO4-,whereas CrO42– begins to dominate for pH > 6. H+ ions adsorbed onto the surface of the composite conducted a positive charge at low pH conditions. Simultaneously, Cr(VI) ions present in different anionic forms are strongly attracted to the positively charged adsorbent surface, further made the Cr(VI) removal efficiency enhanced. Additionally, the passivating layer of Fe(III)-Cr(III) hydroxide compounds on the surface of BC-nZVI@Cell-g-PAA was inhabited to be formatted, lied on the reduction of the degree of corrosion of the nZVI-based material. Therefore, such conditions can better expose effective active sites[43]. On the flip side, a substantial quantity of OH– ions exists in alkaline solutions, which result in a negatively charged adsorbent surface. The subsequent repulsions between BC-nZVI@Cell-g-PAA and Cr(VI) anions lead to reduced adsorption efficiency[44].
Figure 8. Influences of (a) adsorbent dosage, (b) pH, (c) time, (d) initial concentration, (e) temperature, and (f) frequency on Cr(VI) removal by BC-nZVI@Cell-g-PAA.
Issue 8:
Abbreviations should be ellaborated prior to use it first time in the manuscript.
Response:
Thank you very much for your professional comments, we carefully reviewed the manuscript and conducted a thorough revision of the abbreviations, ensuring that all abbreviations are properly expanded upon their first appearance, in accordance with the standards. The necessary changes have been made to the manuscript.
Details:
In Pages 1-2 and 4 of the revised manuscript:
Abstract: A stabilized biochar (BC)–nano-scale zero-valent iron (nZVI) composite (BC-nZVI@Cell-g-PAA) was prepared using cellulose-grafted polyacrylic acid (Cell-g-PAA) as the raw material through in-situ polymerization and liquid-phase reduction methods for the remediation of hexavalent chromium(Cr(VI))Cr(VI)-contaminated water. BC-nZVI@Cell-g-PAA was characterized by XRD, FT-IR, SEM, BET, TEM, XPS. According to the batch experiments, under optimized conditions (Cr(VI) concentration of 50 mg/L, pH=3, and dosage of 2 g/L), the BC-nZVI@Cell-g-PAA composite achieved maximum Cr(VI) removal efficiency (99.69%) within 120 min. Notably, BC, as a carrier, achieved high dispersion of nZVI through its porous structure, effectively preventing particle agglomeration and improving reaction activity. Simultaneously, the functional groups on the surface of Cell-g-PAA provide excellent protection for nZVI, significantly suppressing its oxidative deactivation. Furthermore, the composite effectively reduced Cr(VI) to insoluble trivalent chromium(Cr(III)) species and stabilized them on its surface through immobilization. The synergistic effects of physical adsorption and chemical reduction greatly contributed to the removal efficiency of Cr(VI). Remarkably, the composite exhibited excellent reusability with a removal efficiency of 62.4% after five cycles, demonstrating its potential as a promising material for remediating Cr(VI)-contaminated water.
Along with the industrial activities rapidly expand, contamination of water bodies become severe because of more and more harmful heavy metal released into the aquatic environment. Chromium (Cr) is one kind of heavy metal in the water body environment that has received significant attention based on its extensive usage in various industrial fields, such as mining, electroplating, steel manufacturing, and dye production. Industrial emissions of Cr are a major source of environmental pollution[1], with compounds of hexavalent Cr(VI) being widely recognized for their strong carcinogenic and genotoxic characteristics, posing significant risks to human health. Hexavalent chromium(Cr(VI)) and trivalent chromium(Cr(III)) are main two oxidation states in Cr contamination[2]. Both of them, the former has been given increasing attention according to its great mobility and substantial harmfulness[3], and therefore, requires urgent and effective mitigation strategies.
50 mL aqueous solution containing 4.8 g of cellulose in a 500 mL flask was prepared under mechanical stirred at 90°C for 30 min. The above flask was equipped with a reflux condenser, stirrer, thermometer, and nitrogen line. When the mixture was quenched to 35-40°C, 100 mL n-hexane and 0.12 g of Polyglycerol esters of fatty acids(PGFE)(a suspending stabilizer) was added to and agitated for 15 min. After that, potassium persulfate(K2S2O8) initiator was introduced and the solution was further stirred for 20 min and added with partially neutralized acrylic acid. Then, 2.5 mL of N,N-methylene bisacrylamide cross-linking agent (NN)(5 g/L) were added. The polymerization procession was carried out at 70°C for 3 h. The BC-nZVI powder is then mixed with Cell-g-PAA solution( the mass ratio of BC to Cell-g-PAA is 2:1).
Issue 9:
Please make sure to check typo error through out the manuscript.
Response:
Thanks for this comment. As per your suggestion, we carefully examined the original manuscript for spelling errors and made the necessary corrections.
Details:
In Page 15 of the revised manuscript:
BC-nZVI@Cell-g-PAA was treated with Alkali solution and acid solution after eh.ach repeated experiment, and an amount of BC-nZVI@Cell-g-PAA was lost in the cleaning process[50].
Issue 10:
N2 adsorption desoprtion and particle size distribution plots should be incorporated in the manuscript with respective section.
Response:
Thank you very much for your professional comments. We sincerely apologize for the omission of the adsorption-desorption curve and pore size distribution plot in the original manuscript. Currently, we have completed the addition of the relevant content.
Details:
In Page 13 of the revised manuscript:
Figure 7. (a) N2 adsorption-desorption isotherm diagram; (b) aperture profile.
Issue 11:
Can Cell-g-PAA coating layer thickness have any effect over the chromium removal performance of the catalyst?
Response:
Thank you very much for your professional comments. Based on our experimental results, the variation in coating thickness does indeed affect the material's reaction performance. We have discussed in detail the impact of coating thickness on the material's performance and have supplemented with corresponding experimental data and analysis to better address your question.
Details:
In Page 8 of the revised manuscript:
HRTEM provides an intuitive way to observe the distributn of Fe particles on the BC-nZVI@Cell-g-PAA composite surface, along with key characteristics, such as particle uniformity, size, crystal facet exposure, and interplanar spacing (Figure 4). The dispersant of Fe nanoparticles on the BC surface was uniform and the average size of these articles was 9.8 nm. This uniform dispersion of Fe was relayed on the relatively large surface area of BC. Furtherly, lattice spacing of the (110) and (200) crystal facet positions were 0.246 and 0.273 nm, respectively[25].The thickness of the Cell-g-PAA coating layer does affect the Cr(VI) removal performance. Specifically, as the Cell-g-PAA coating ratio increases, the reduction ability of Cr(VI) first improves and then decreases. The improvement is attributed to two main reasons: (1) the larger the molecular weight of Cell-g-PAA, the stronger the electrostatic repulsion effect it provides, leading to smaller and more stable nZVI particles with a higher surface area, thus enhancing reactivity; (2) Cell-g-PAA effectively isolates nZVI from oxygen, preventing premature oxidation. However, when the coating layer becomes too thick, it significantly increases the viscosity of the nZVI slurry, causing a reduction in the reduction ability, as well as difficulties in transfer and drying. The relevant image is shown in Figure S1 of the supporting materials.
Supporting material:
Figure S1. Effect of coating ratio on Cr(VI) removal.
Special response to Editorial Office and reviewers:
The numbers of some sections in results and discussion, figures, tables, and references have been renumbered for the reorganization of the present manuscript. We are sorry for any inconvenience caused by that. At last, we would like to thank you sincerely for your valuable suggestions and efforts that have been made during the review.

Round 2
Reviewer 1 Report
Comments and Suggestions for Authors
Accepted in current form
Comments on the Quality of English LanguageImproved the quality of English
Author Response
Dear reviewer,
Thanks very much for your kind work and consideration on publication of our paper. On behalf of my co-authors, we would like to express our great appreciation to editor and reviewers.
Thank you and best regards.
Yours sincerely,
Yan Zhang
Reviewer 2 Report
Comments and Suggestions for Authors
Recommended for publication
Author Response

(The authors gave the same response as above.)
